# Adaptations by the coral *Acropora tenuis* confer resilience to future thermal stress

Sanaz Hazraty-Kari [1✉], Parviz Tavakoli-Kolour [1], Seiya Kitanobo[2], Takashi Nakamura[1,3] & Masaya Morita [3✉]

Elevated temperatures cause coral bleaching and reef degradation. However, coral may have strategies to survive by reproducing more heat-tolerable larvae. We examine the direct and carryover effects of thermal stress on fecundity and fitness in the reef-building coral *Acropora tenuis*. Fragments from the same colony are subjected to control temperature (~27.5 °C) or heat stress (~31 °C) for ten days. We then examine the fecundity of adults (egg number and size) and the thermal tolerance of larvae and recruits (survival rates, growth, and size). The stressed fragments show a trade-off in egg production, an increase in egg number but a decrease in size. In addition, larvae and recruits from the stressed colony show marginally higher survival rates in the higher water temperature but do not differ in the control condition. Therefore, corals produce more heat-resistant larvae and recruits after they experience heat stress, which may improve coral reef resilience.

[1] Graduate School of Engineering and Science, University of the Ryukyus, Okinawa, Japan. [2] Shimoda Marin Research Center, University of Tsukuba, Shimoda, Shizuoka, Japan. [3] Sesoko Station, Tropical Biosphere Research Center, University of the Ryukyus, Okinawa, Japan. ✉email: s.hazrati@hotmail.com; morita@lab.u-ryukyu.ac.jp

Global warming represents a widespread threat to a variety of communities, including coral reefs[1,2]. In recent decades, coral reef communities worldwide have experienced mass bleaching and mortality owing to temperature anomalies[3,4], leading to reef decline[5]. Coral bleaching was especially heavy from 2014 to 2017[4], and more than 70% of coral reefs worldwide have experienced heat stress due to a temperature increase of 1–2 °C[4]. Severe bleaching is expected to become ubiquitous within the next few decades[1,6], leading to the potential collapse of coral reefs[5,7,8].

Recent findings indicate that corals potentially acclimate to the higher water temperature without bleaching and reduction of physiological activity such as photosynthesis, respiration, growth, and survival rate[9–12]. For example, many species such as massive corals *Porites lutea*, are heat-tolerant and can survive thermal stress in nature[13]. On the other hand, most *Acropora* spp. cannot survive[13]. The differences in the responses against the higher water temperature are due to species and durations of stresses.

The higher water temperature caused several negative impacts on coral physiology[14,15]. For example, higher water temperature declines physiological performance such as respiration[16,17] and photosynthetic activity ($Fv/Fm$)[16], reproduction[18,19], number and size of eggs[20,21], larval settlement[22], and recruitment[23,24]. These physiological parameters could reduce the capacity for coral reef recovery[18] and influencing the population dynamics of the ecosystem[25–27].

The adult reproductive output as measured by egg production varies after thermal stress. For many coral taxa, the effect of heat stress reduces the number of eggs[21,28]. In addition, the fecundity of the colony also declines due to the smaller number of polyps producing eggs[28]. Although thermal stress is implicated in changes in egg sizes, the trade-off between eggs size and fecundity has not been clearly shown[29]. Moreover, the inheritance of larval traits from the stressed colony are not well documented.

The acclimation potential of coral species at different life stages from larval and juvenile to adult has not been evaluated systematically. For example, larval stages are short (<2 months), but the juvenile stages after settlement until reproduction are longer (>5 years). From spawning to fertilization up to swimming larvae, higher water temperature (>31 °C) causes a marginal reduction in development at the gastrulation stage in several coral *Acropora* species[26,30,31]. In terms of larval survival, water temperature influences respiration, photosynthetic activity, and settlement rates, but the degree vary among species[16,17,26,31–34]. In addition, high water temperature (29 to 32 °C) does not consistently cause negative impacts, such as mortality, on their larval stages in several species (mostly <48 h). In contrast, the more prolonged exposure to the high temperature (>150 h) after settlement decreases survival rates and photosynthetic activity in the recruits[16,17,31–34]. The differences in the reaction at the larval and juvenile stages may be due to the duration of the exposure to thermal stress, but we need to clarify it. In addition, larvae from the higher water temperature show more rapid respiration, implying the acclimation potential of the larvae when the adults experience thermal stress[17].

Although the early life stages are crucial for the life cycles of the corals, the variations of the reactions against heat stress at early life stages are not consistently examined between larvae from stressed- and non-stressed adults. Investigations in this regard may reveal critical ecological issues, especially in cases where tolerances vary across life cycles due to future global warming. The early life stages of reef-building corals are crucial for reef maintenance and the replenishment and persistence of degraded coral reefs[35,36]. These life stages are also affected by surrounding water temperatures. The thermal stress history of parents can influence the thermotolerance of early life stages in corals and

other marine organisms[17,37]. This study intended to reveal how consecutive stress on adult corals affects larval and juvenile performance under similar stress conditions.

The coral species *Acropora tenuis* inhabits shallow reefs with temperature fluctuations, and these fluctuating temperatures may affect both larval settlement and juveniles[38]. Responses to temperature (such as larval recruitment) also vary across species[39], and the effects of thermal stress on adults, larvae, and juveniles have been reported in previous studies[17,22,31,40]. In a higher water temperature, larval development becomes faster[22,31]. In addition, metamorphosis leads to settlement occurring faster[41] but also shows higher mortality[22,31,34]. As a result of the more rapid development of the larval stages, they tend to settle near the adult colony[42]. The larval dispersal becomes smaller, and the number of recruited larvae could increase after the degradation of the reef due to heavy bleaching events. From this aspect, more heat-tolerant larvae and recruits are desirable in a higher water temperature and will be a good strategy for their fitness after colonies experience heat stress. However, the effects of successive thermal stress on adults and the subsequent impact on their fecundity, the larval stage, and the juveniles have not been examined.

In this study, we hypothesized that corals might have a tactic to acclimate to increase their fitness, increasing survival rates in higher temperatures. Then, we examined the consecutive effects of heat stress on the reproduction and early life stages (oocytes, larvae, larval settlement, and early juveniles) of corals (Fig. 1). We applied the thermal stress to adult colonies and observed how their egg production changes, and the degree of thermal tolerance of the larvae and recruits. The adult colonies were divided into two fragments to exclude the effect of genetic background. They compared the stress responses from the adult stages, their egg production, larvae, and the recruits. The control temperatures were ambient (26 to 28 °C), and higher water temperatures were around 31 °C.

## Results

**Physiological responses of adult colony fragment to thermal stress**. First, we measured the brightness of the coral fragments, a color scale corresponding to symbiont loss. Brightness (color scale) was significantly higher in stressed fragments (SF) than in control fragments (CF) (GLMM $p < 0.0001$, Fig. 2a). The dark-adapted quantum yield of photosystem II ($Fv/Fm$) was significantly lower in SF than in CF, as estimated using a linear mixed model (LMM, $F = 27.65$, $p < 0.0001$). There was an apparent reduction in $Fv/Fm$ from day 5 of exposure to elevated temperatures. In contrast, CF maintained stable $Fv/Fm$ values during the experiment (Fig. 2b).

Symbiotic algal density was measured in the SF and CF in all treatments before and after thermal stress. SF lost >35% of their initial algal cell density (Fig. 2c), whereas CF ($11.71 \times 10^5 \pm 1.15$ cells cm$^{-2}$; mean ± standard error of the mean SE) exhibited higher algal density than SF ($4.33 \times 10^5 \pm 0.82$ cells cm$^{-2}$) at the end of experiment. After 10 days of exposure to elevated temperatures, there was a significant difference in algal density between the treatments (unpaired two-sample $t$-test, $p = 0.0003$). Chlorophyll-$a$ and -$c_2$ contents per surface area decreased in SF during the experimental period. Although the chlorophyll-$a$ content per surface area was not significantly different between groups (unpaired two-sample $t$-test, $p = 0.09$), chlorophyll $c_2$ levels were significantly lower in SF than in CF (unpaired two-sample $t$-test, $p = 0.02$) (Fig. 2d, e).

One fragment from the stressed treatment died three months after the thermal stress. Survivorship was not significantly different between CF and SF (Kaplan–Meier log-rank test, $p = 0.3$). The survival percentage was 100% in CF but 83.3 % in SF (Fig. 2f).

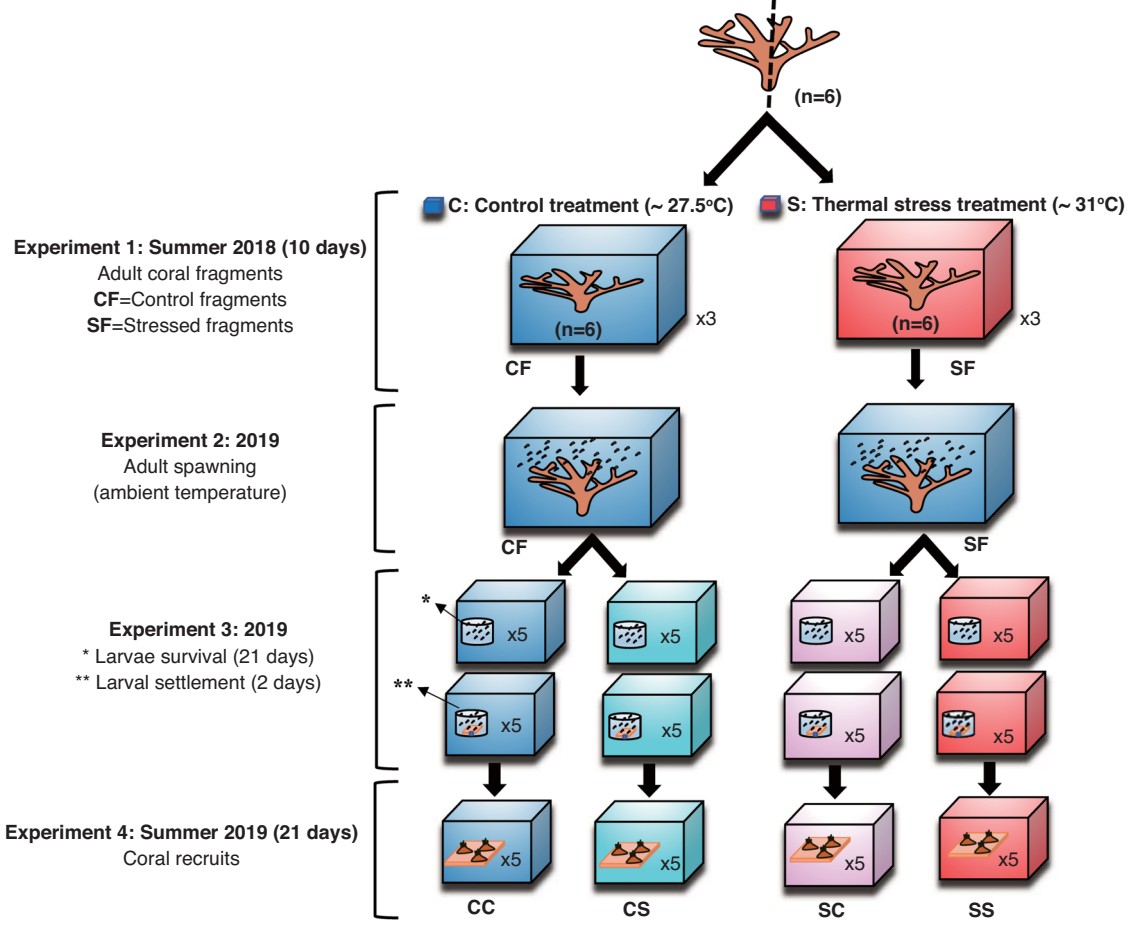

**Fig. 1 Experimental design to examine the carryover effects of thermal stress on fecundity in adults and thermal tolerance in larvae and recruits of _Acropora tenuis_.** Six colonies of _Acropora tenuis_ were sectioned into two fragments and divided into two groups. The fragments were exposed to two temperature treatments: 1. thermal stress at ~31.2 °C (red aquariums containing stress fragments); and 2. control temperature at ~27.5 °C (blue aquariums containing control fragments). The four experiments were as follows: Experiment 1. The stress treatment group (SF) was exposed to thermal stress for 10 days. The Control group (CF) was also reared in the control condition for 10 days. Experiment 2. Effects of thermal stress on fecundity in adult corals. Experiment 3. Effects of thermal stress on the larval stage and larval settlement. Experiment 4. Effects of thermal stress on recruits obtained from stressed and control fragments. CC indicates larvae or recruits from the control fragments (CF) in the control condition, CS is larvae or recruits from CF in the stressed condition, SC is larvae or recruits from the stressed fragments (SF) in the control condition, and SS is larvae or recruits from SF in the stressed condition.

**Gamete characteristics of adults.** Four colonies in stressed (SF) and six of those in the control treatment (CF) spawned. One colony of the stressed condition (SF) did not spawn. The number of eggs per bundle increased for SF replicates (10.79 ± 0.48) than in CF (9.08 ± 0.36) (Fig. 3a), and the difference was statistically significant (unpaired two-sample Wilcoxon test, $p = 0.003$). In contrast, egg volume decreased significantly after thermal stress. The mean egg volume was significantly (unpaired two-sample Wilcoxon test, $p = 0.0002$) higher in CF (0.7 ± 0.04 μm³; range, 0.52–1.17 μm³) than in SF (0.52 ± 0.02 μm³; range, 0.34–0.83 μm³) (Fig. 3b).

**Larval responses to thermal stress.** We examined size, lipid content, and survival rates of the larvae of each treatment. Larvae size in CF (1087.82 ± 51.58 μm) were larger than SF larvae (868.33 ± 19.16 μm), and the difference was significant (unpaired two-sampled Wilcoxon test, $p = 0.00003$; Fig. 3c). Lipid depletion rates in 5-d-old _A. tenuis_ larvae varied among the treatments (SF and CF larvae). Lipid contents were significantly lower in SF larvae than in CF larvae (linear model (LM), $p = 0.02$). On average, the initial lipid contents were 13.72 ± 0.36 μg in SF larvae

and 17.33 ± 0.57 μg in CF larvae (Fig. 3d). After 21 days of exposure to control and thermal stress conditions, the lipid contents of the larvae decreased significantly in all treatments (LM, $p < 0.01$) (Fig. 3d). Lipid content was lower in larvae exposed to thermal stress than in those exposed to control treatments; however, there was no significant difference between the groups. Lipid depletion rates were highest in the control treatments (CC, 32.3%; SC, 29.62%) and relatively lower in the stress treatments (CS, 23.91%; SS, 19.8%).

Survival rates of the larvae were different between CF and SF treatments. During the 21 days of thermal stress treatment, survival rates were lowest in CF larvae under thermal stress (CS; 82.6 ± 1.6%) and SF larvae under thermal stress (SS; 85.0 ± 1.5%). Survivorship was not significantly different between CS and SS (Kaplan–Meier log-rank test, $p = 0.08$). In contrast, survival rates were higher in CF larvae under control conditions (CC; 92.6 ± 1.29%) and SF larvae under control conditions (SC; 91.4 ± 0.75%). Survival rates were not significantly different between CC and SC (Fig. 3e). However, the results of the log-rank test showed that survival rates differed significantly between larvae in high-temperature and control conditions (CC-CS, SC-SS) (Supplementary Table 1).

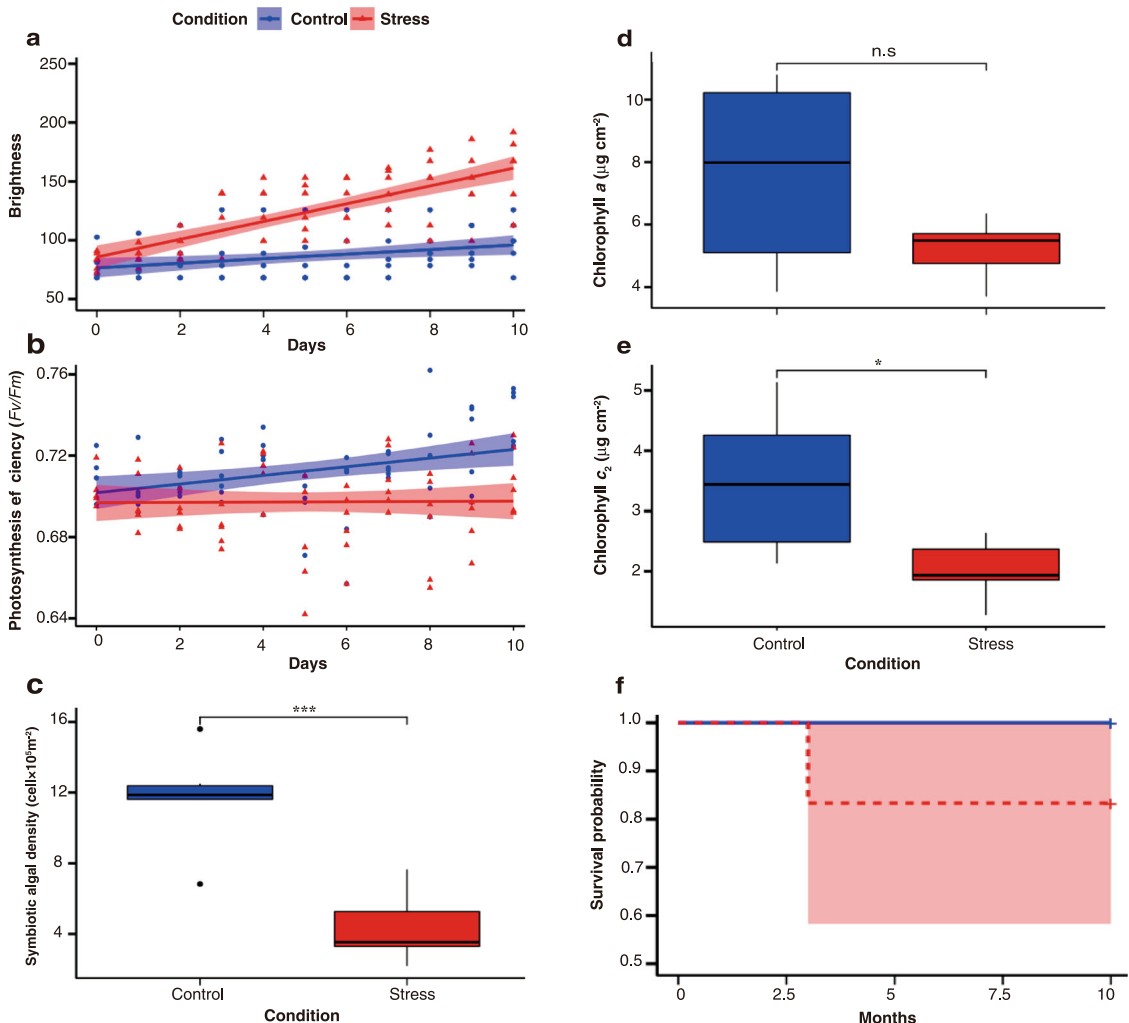

**Fig. 2 Effect of thermal stress on the physiological performance and survival of the adult fragments of *Acropora tenuis* (Experiment 1). a** Brightness (color scale) of the fragments exposed to thermal stress (red line and red triangles) or in control conditions (blue line and blue circles). *y*-axis values of 50 and 255 are equivalent to 100% health and bleaching status, respectively[64]; (*n* = 6 for each treatment). Shading indicates 95% CI. **b** Changes in the maximum quantum yield (*Fv/Fm*) following exposure to thermal stress and control conditions. The lines are fitted by a linear mixed model (*F* = 27.65, *p* < 0.0001; *n* = 6 for each treatment). Shading indicates 95% CI. **c** Algal symbiont densities after 10 days. Algal densities were significantly different between the control (blue) and stress (red) fragments (unpaired two-sample *t*-test, *p* = 0.0003; *n* = 6 for each treatment). **d** Chlorophyll-*a* after 10 days. The chlorophyll-*a* content was not significantly different between the control (blue) and thermal stress (red) groups (unpaired two-sample *t*-test, *p* = 0.09) and **e** Chlorophyll *c₂* contents of corals after 10 days significantly different between the control (blue) and thermal stress (red) groups (unpaired two-sample *t*-test, *p* = 0.02). **f** Mean survival (%) of coral fragments exposed to thermal stress (red) and control conditions (blue). Coral fragments in the two temperature treatment groups were significantly different survival rates (*p* = 0.03 by the Kaplan–Meier log-rank test; *n* = 6 for each treatment). The boxplots show the median, interquartile range, and spread of data. Asterisk on the top of each box plot (*p*-values) indicates a significant difference between treatments (*** refers to *p* < 0.001, ** refers *p* < 0.01 and * refers to *p* < 0.05). n.s refers to no significant. Shading indicates 95% CI.

**Effect of thermal stress on settlement rate and size.** On day 1 of the larval settlement period, larval settlement rates were faster in thermal stress than in control treatments for larvae from both stressed and control fragments. The cumulative settlement rates in CS (82.6 ± 1.54%) were higher than those in the other treatments. The settlement rate was lowest in SS (69.2 ± 8.17%). Among the control treatments, SC larvae (81.8 ± 3.31%) exhibited a higher settlement rate than CC larvae (74.4 ± 2.14%). Overall, the cumulative number of settled larvae did not differ significantly among treatments (pairwise Wilcoxon test, *p* > 0.05) over the settlement period (Fig. 4a; Supplementary Table 3). Early settlement sizes in the control treatments were larger than in the stress treatments. In particular, settlement sizes in CC were significantly larger than those in the corresponding stress treatments, including CS (LM, *p* < 0.0001), SC (LM, *p* < 0.004), and SS

(LM, *p* < 0.0001) (Fig. 4b). The settlement sizes did not differ significantly between the SC and SS groups (Fig. 4c LM, *p* = 0.58; Supplementary Table 4).

**Effect of thermal stress on post-settlement survival.** After 21 days of exposure to thermal stress, the survival rates of recruits were highest in the control treatments, including SC (87.04 ± 4.74%) and CC (85.46 ± 2.53%). The survivorship of recruits was significantly lower in stress treatments compared with that in the corresponding control treatments (Kaplan–Meier log-rank test; CS vs. CC, *p* < 0.0001; SS vs. SC, *p* < 0.0001) (Supplementary Table 2). However, recruit survival rates were significantly higher in SS (72.31 ± 10.77%) than in CS (66.2 ± 12.45%) (Kaplan–Meier log-rank test, *p* < 0.0001).

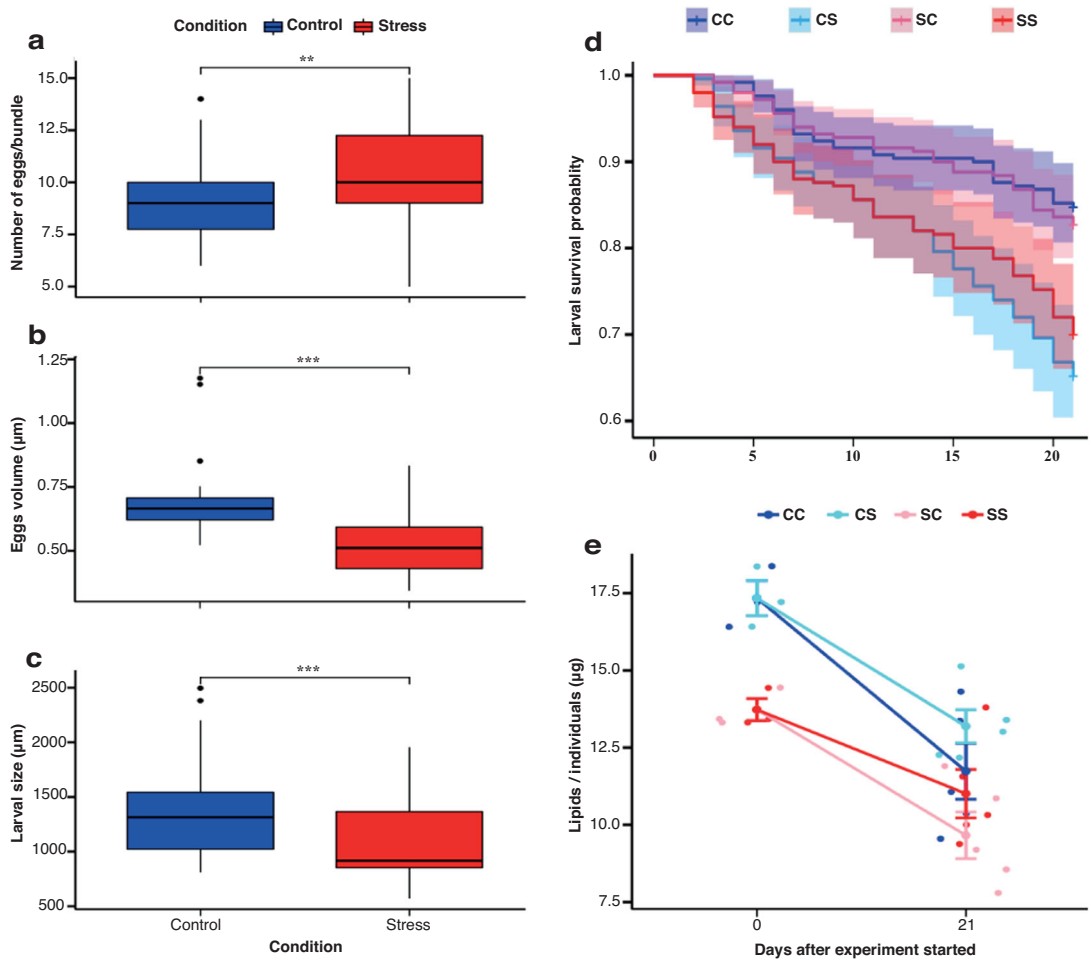

**Fig. 3 Carryover effects of thermal stress on fecundity of *Acropora tenuis* (Experiment 2). a** Number of eggs in the bundles of control (blue) and stressed (red) fragments. The average number of eggs for 6 bundles in each treatment group is shown (stress; $n = 24$ bundles from 4 colonies, control; $n = 30$ bundles from 5 colonies). The values differ significantly between treatments (unpaired two-sampled Wilcoxon test, $p = 0.003$). **b** Egg volume from control (blue) and stressed (red) fragments. The egg volume in bundles is significantly different between the stressed (red, $n = 20$) and control (blue, $n = 20$) fragments (unpaired two-sampled Wilcoxon test, $p = 0.0002$). **c** Mean larvae sizes (5 days post-fertilization) from gametes in the control (blue) and stressed (red) treatment groups. Larval sizes differ significantly between stressed fragments (SF, red; $n = 45$) and control fragments (CF, blue; $n = 69$) (unpaired two-sample Wilcoxon test, $p = 0.00003$). **d** Larval survival (%) in the two treatment groups ($n = 100$ per treatment) exposed to control conditions (SC, pink line; CC, dark blue line) and thermal stress (SS, red line; CS: cyan line). The survival curves show the effect of thermal stress on larval survival in the control and thermal stress conditions over 21 days (Kaplan–Meier log-rank test, Supplementary Table 1). The survival curves of larvae under thermal stress (CS and SS) are not different between treatments (Kaplan–Meier log-rank test, $p = 0.37$). Shading indicates 95% CI. **e** Lipid consumption rates in larvae obtained from the gametes of control fragments (CC: dark blue, CS: cyan) and stressed fragments (SS: red, SC: pink) at different temperatures. In the all larval experiments, the control conditions were ambient temperature in early summer. The water temperatures changed within a day and during experimental days. Error bars indicate standard errors (S.E.). CC indicates larvae or recruits from the control fragments (CF) in the control condition, CS is larvae or recruits from CF in the stressed condition, SC is larvae or recruits from the stressed fragments (SF) in the control condition, and SS is larvae or recruits from SF in the stressed condition.

## Discussion

The present study shows that coral colonies under thermal stress can produce more eggs, and the larvae showed a slight heat tolerance. Thermal stress can have effects on egg size and fecundity[43,44]. In the present study, egg volumes were smaller in SF treatments than in CF treatments (Fig. 3b), but the number of eggs per bundle was significantly higher in SF than in CF. The reduction of eggs and larvae indicates a relationship between egg size and egg number (fecundity) in corals[45,46]. In other words, larger eggs result in a reduced number of gametes[47].

In short-term thermal experiments on adults, the fragments exposed to thermal stress (SF) have shown a change in physiological parameters, such as a reduction in symbiotic algal density, chlorophyll concentration, photosynthesis efficiency, and an

increase in brightness compared to those control fragments (CF). It follows that photochemical efficiency (*Fv/Fm*) decreases[48], which is one of the first responses to thermal stress[49–51]. Although thermal stress affected fragments, five colonies were recovered within ten months in the reef, and only one colony could not survive. One did not spawn in the spawning season, which can be due to differences in the thermal capacity of colonies. The negative impact of thermal stress on reproductive outputs, including the absence of eggs[28] and reduction in fecundity, has been found across many coral species[21,52,53].

Thermal stress in the adult stage affected gamete production (producing more but smaller eggs), but heavy bleaching may cause different consequences (abandoning reproduction). As described in previous studies[29,43], heat stress causes the fragments

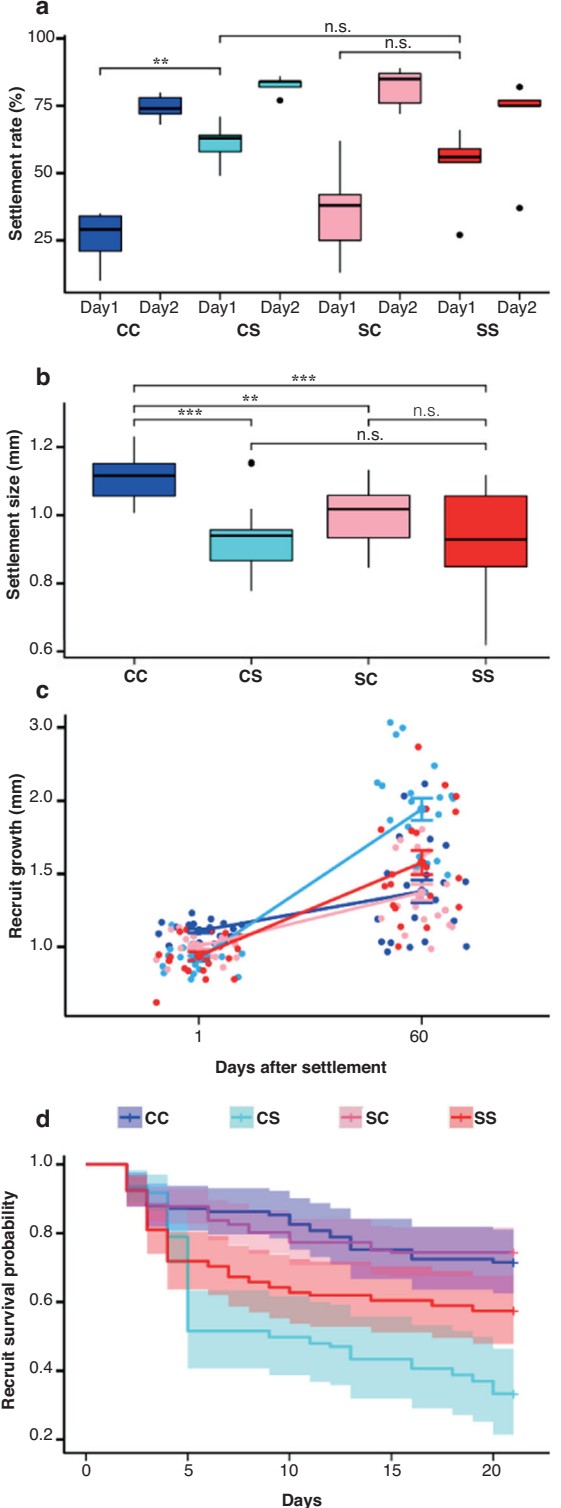

**Fig. 4 Effect of thermal stress on the larval settlement rates and size (Experiment 3), and the recruit growth and survival rates (Experiment 4) of *Acropora tenuis*. a** Effect of thermal stress on larval settlement rates in control (CC and CS) and thermal stress (SC and SS) treatment groups. Settlement rates are shown after 24 h (1 day) and 48 h (2 days) under high (~31 °C) and control (~28 °C) temperatures ($n = 100$ larvae per treatment) after 24 and 48 h from the beginning of the experiment. Each box indicates CC (Dark blue), CS (cyan), SC (pink), and SS (red). **b** Settlement size (mm) in each treatment group. The largest diameter of the recruits was measured as "size". ($n = 20$ recruits in each condition, CC, CS, SC, and SS). Each box indicates CC (Dark blue), CS (cyan), SC (pink), and SS (red). There are no significant differences between the two treatments (LM; $p > 0.05$). **c** Recruit growth 60 days after settlement. Growth rates between CS and SS were significantly different (LM $p < 0.009$; Supplementary Table 5), but those between CC and SC were not (LM $p = 0.7$; Supplementary Table 5). Error bars indicate standard errors (S.E.). **d** Survival probability of the recruits from larvae of the stressed or control treatment in control and higher water temperature. The recruits were exposed to high and control conditions for 21 days. The probability between CS and SS was significantly different (Kaplan–Meier log-rank test $p < 0.05$, Supplementary Table 6) but that of CC and CS was not (Kaplan–Meier log-rank test $p = 0.67$, Supplementary Table 6). Shading indicates 95% CI. CC indicates larvae or recruits from the control fragments (CF) in the control condition, CS is larvae or recruits from CF in the stressed condition, SC is larvae or recruits from the stressed fragments (SF) in the control condition, and SS is larvae or recruits from SF in the stressed condition.

*purpurea*. The heat-treatment of the adult colonies causes an increase in the number of larvae, but their size is smaller[54]. The *Lepstastrea* is not phylogenetically relative to the genus *Acropora*, but this thermal response is similar; A change in larval number due to an increase in egg number. Corals may abandon investing energy in gamete production when they are stressed, but they produce more eggs when they are slightly stressed. There may be a range of length and temperature of heatwaves, and we need to specify this range.

Regardless of thermal stress, the lipid content of the larvae is supposed to differ with egg size. Although we did not measure the lipid content of the eggs, the smaller lipid content of the larvae may represent the content of the lipid in the eggs. In addition, the reduced lipid content of larvae did not affect their survival rate. In contrast, other studies have suggested that lipid levels in larvae may be associated with energy allocation in larvae[55] and the sizes of eggs and larvae[44], indicating that smaller eggs would have fewer energetic reserves (lipids) than larger eggs and larvae[20,43,44]. Nevertheless, in this study, the lipid content of the larvae was positively correlated with their survival rates, and the rate of lipid consumption was lower in SF larvae than in CF larvae under thermal stress after 21 days. The stressed coral fragments produced smaller larvae, slightly tolerant to thermal stress.

After dispersal with lower lipid levels, the larvae and recruits were slightly tolerant to high water temperatures. There were no significant differences in the larval settlement between SF and CF larvae. However, larval development and metamorphosis utilize energy from lipids[56,57]. In contrast to our findings, previous studies have reported adverse effects of high temperature on larval settlement[14,58]. Although we did not examine symbiotic- or aposymbiotic larvae throughout the experimental period, we used filtered seawater that may not contain algae. Therefore, most larvae in this study may be aposymbiotic. In our previous study, the symbiotic relationship affects lipid consumption, and the survival rates of the aposymbiotic larvae are less than those of symbiotic[55,59,60]. Although our study shows in cases of low-lipids aposymbiotic larvae, the larvae could build symbiotic

to produce smaller eggs, but the egg number in the bundles becomes larger. On the other hand, corals may abandon producing eggs when they are bleached[43]. The fecundity of the colony also declines due to the smaller number of polyps producing eggs[43]. The differences in the results may be due to the duration of the stress. In this study, we applied a marginally shorter duration of heat stress, but apparent bleaching did not occur. The colony might respond to the thermal stresses and switch to producing more eggs, as shown in the previous study[29]. This trade-off is also observed in the brooding corals *Lepstastrea*

relationships in nature. Therefore, the survival rates of the low-lipid larvae from SF parents could increase.

Higher water temperature influences lipid consumption rates but may not be largely related to the growth in the recruits to juveniles. In this study, the lipid consumption rates decreased at higher temperatures in the larvae and recruits. The declines of lipid consumption in larvae and recruits from both control and stressed fragments contradict with rapid development, growth rates, and settlement of the larvae. Lipid consumption rates in *P. damicornis* becomes slower at higher water temperature[59]. However, the slower lipid consumption rates in the larval stage can be beneficial for the extension of larval longevity[61]. The lipid consumption rates change according to the water temperature, but further study is required to elucidate the relationship between physiological function and the rate of lipid consumption rates.

The size of the recruits could be correlated with the survival and growth rates in the corals, but the slight differences are supposed to be not critical for their survival. Although the size of the recruits from the stress fragments was smaller, the survival rates of the smaller recruits from the stress fragments in higher water temperatures were higher (SS Fig. 4d). The size of the recruits to juveniles affects their survival rates. The larger size of the recruits shows higher survival rates in the brooding coral *Pocillopora damicornis*[40]. The contradiction of the size and survivorship in this study, smaller recruits showing more heat tolerance, implies reactions to the thermal stress could be varied, and responses are under complex physiological reactions.

Larvae and recruits from the stressed fragments showed slight heat tolerance compared to those from non-stressed control, implying that higher survival at larval to recruit stages in higher temperature is slightly beneficial for reef maintenance/resilience in the warming ocean. In this study, larvae showed more rapid settlement rates at higher water temperature (Fig. 4b CS and SS). As in our study, the higher water temperature causes restriction of larval dispersal due to rapid larval development and settlement[22,31,41,42]. Therefore, the larvae need to grow near the habitat of the parent colonies, and the water temperature in the summer of their habitat potentially becomes higher. In the case of the *Acropora tenuis*, their habitat is shallow (<10 m) and primarily influenced by heatwaves in the summer[62]. Although the frequencies of the heatwaves are not considered in this study (exposed once 10 days a year), the stressed colony could produce more slightly heat-tolerant larvae. This acclimation potential is helpful for their population. Still, reproductive success after spawning is limited when the number of colonies declines due to heavy bleaching. In addition, the juveniles could not survive when the summer heatwaves frequency was high (Hazrati et al., in preparation).

In conclusion, this study showed that thermal stress influenced reproduction and the early life stages of corals without causing any changes in energy investment. We also found that colonies that experienced heat stress developed tactics to increase larvae and recruit survival in higher water temperatures. We set ambient temperature as the control condition in the early summer; thus, the water temperature was unstable and varied by about 2 °C (26 to 28 °C). However, the higher water temperature was stable within 1 °C difference (31 °C). There were at least 3-degree differences. Therefore, if parents experience the heatwaves, they may prepare eggs to survive their larvae in heatwaves, which could arise next spawning. However, further research is required to understand the relationship between warm ocean temperatures, coral recruitment survival, and recovery potential over successive generations.

## Methods

**Coral collection and experimental design.** In this study, we used the reef-building coral *Acropora tenuis*. The coral *Acropora* is a broadcast spawner. Larvae disperse after fertilization within released gametes and recruit in the reef. Here, we examined the effects of thermal stress on the following life stages and functions in the coral *Acropora tenuis*: adults physiology (Experiment 1), adult fecundity (egg number and size) (Experiment 2), larval survival and settlement rates (Experiment 3), and recruits (Experiment 4) (Fig. 1). In larval and recruits experiments, we classified the following conditions: larvae from control colonies exposed to control condition (CC), larvae from control colonies exposed to stressed condition (CS), larvae from stressed colonies exposed to control condition (SC), larvae from stressed colonies exposed to stressed condition (SS).

In Experiment 1, six colonies of the *Acropora tenuis* were collected from five to seven m depth in the south of Sesoko Island (26° 38' N, 127° 52' E). These colonies were transferred to the Sesoko Station, Tropical Biosphere Research Center of the University of the Ryukyus, located in southern Japan in August 2018. All colonies were sectioned into two fragments and were tagged as two different treatments (control and stress treatments). To examine thermal stress and its carryover effects, we used two coral fragments from the same colony to ensure a uniform genetic background and used six colonies for making fragments (Fig. 1 and Supplementary Fig. 1). Twelve coral fragments were exposed to two temperatures—27.5 ± 0.02 °C on average (control) and 31.2 ± 0.01 °C (stress)—for 10 days in a flow-through system (see below). In the stress experiment, we placed a heater in each tank, and the temperature was gradually increased from 27.5 °C to ~31.2 °C over the course of 1.5 days (~0.5 °C every 6 h). In heat stress conditions, we applied +2 degrees for 10 days of stress to the fragments. This heat duration was equivalent to 20 °C heating days (DHD)[63], (2 °C above the average maximum monthly temperature) for 10 days. The control condition corresponds to the average water temperature in the shallow reef in the summer. All fragments were maintained in an outdoor tank for 10 days in flow-through seawater under natural solar irradiation before starting the experiment (acclimation period). After acclimation, we randomly placed fragments in the three replicate tanks as control and stress conditions. These tanks were equipped with a flow-through system. The water temperature conditions were recorded every ten min using a temperature logger (HOBO Water Temp Pro v2, Onset) for 10 days. The water flow was supplied using two water pumps at the opposite corners to ensure the water's mixture within each tank. During this experiment, the fragments were maintained under 12-h light (160 μmol quanta $m^{-2} s^{-1}$) and 12-h dark conditions (Supplementary Fig. 1a).

We daily photographed the fragments to monitor their color (brightness) and the maximum quantum yield of photosystem II (*Fv/Fm*) throughout the experiment (see below). We also measured their physiological parameters (color, survival provability, the symbiotic algal density, and chlorophyll content) before and after the experiment (see below). Finally, all fragments were transplanted back to the reef until spawning occurred the following year (Supplementary Fig. 1).

### Physiological parameters of coral adults

*Coral color measurements (brightness).* The fragments of each treatment were photographed daily using a Canon Powershot G10 digital camera with a constant white balance and under identical illumination and Coral Health Chart (www.coralwatch.org) for monitoring coral health and color. Coral Health Chart was used to indicate the degree of health, paling, or bleaching as a reliable proxy for estimating the changes in symbiont density and chlorophyll-*a* and -*c2* content. The D hue was applied for *A. tenuis*, which consisted of six different colored areas (D1, D2, D3, D4, D5, D6); D1 (white) shows the lightest and D6 (brown) displays the unbleached color. The color reflection of the three-part of the fragment in the images was analyzed by histogram function using Adobe Photoshop CC 2015 and matched with the coral health chart D hue category. The color reaction values ranged from 50 to 255, equivalent to 100% health and bleaching status, respectively. Specific details of this parameter have been described in Siebeck et al.[64].

*The maximum photosynthetic quantum yield (Fv/Fm).* We evaluated the photosynthetic activity of coral fragments in each treatment group by measuring the maximum quantum efficiency of photosystem II (*Fv/Fm*). We measured one hour after the sunset flowing 60 min in the dark-adapted state at three locations of the fragments throughout the experiment and using diving pulse amplitude modulated (PAM) underwater fluorometer (Walz, Effeltrich, Germany). The measurements were repeated daily over 10 days experiment period.

*Determination of symbiotic algal density and chlorophyll-a and -c2.* In each treatment, one branch of each fragment was randomly harvested at the beginning and end of the thermal stress experiments to analyze symbiotic algal density and chlorophyll (Chl)-*a* and -*c2* content. Only one branch was collected to avoid escalating stress and keep the fragments surviving until the subsequent spawning in 2019. The samples were frozen at −80 °C until further processing and defrosted, then their tissues were removed from each branch of the fragment with an air-Pik and filtered seawater applied inside a Ziploc bag. After homogenization, the homogenate was centrifuged at 4000 g for 20 min at 4 °C) and washed 3 times. The washed homogenates were divided into two aliquots. We immediately counted symbiotic algae in the first aliquot of five drops as replicates using a hemacytometer

under a light microscope (Olympus, Tokyo, Japan). Finally, following the standard procedure, the algae's mean total number per mean surface area of each fragment was divided into treatments based on the paraffin wax dipping technique[65]. One milliliter of the second aliquots was centrifuged ($1000 \times g$ for 15 min at 4 °C) and the pellets were used to extract the chlorophyll-*a* and -*c*2 concentrations. The pellet was mixed with one ml of 90% acetone and was quantified after 24 h in the dark at 4 °C. To calculate the chlorophyll content, the extract solutions were measured in the different path lengths described by Jeffrey and Humphrey[66] with a spectro-photometer standardized by (Hitachi U-2001, Japan) to the surface area of the branch. These parameters were determined as a metric tool for coral bleaching.

*Observation of fragment survival.* Following the thermal stress experiment, the fragments were transferred to the reef, and their recovery status was monitored. The fragments were photographed monthly using an underwater camera (Canon Powershot G10, Canon Inc., Tokyo, Japan). Survival rates were recorded 10 months after the thermal stress experiment.

*Experimental design.* In Experiments 2, 3 and 4: In June 2019, we examined the effects of thermal stress on larval stages, larval settlement and size, survival recruits, and growth in *A. tenuis* (Fig. 1). One week before the anticipated spawning date (June 5), fragments from the previous year's thermal stress experiment (CF and CS) were retrieved from the reef. They were transported to an outdoor tank supplied with a flow-through seawater system and kept there until spawning. During the spawning time, the fragments were put into the individual aquarium (one fragment per aquarium, supplementary Fig. 1c). Five fragments from stress treatments were alive four of them were spawned at 19:30–20:40 h on June 12, 2019, around the day of the full moon and one SF did not spawn at all. The gametes were gently scooped from the water surface and transferred to the laboratory. We then mixed gametes from the same treatment groups (not other treatments) for cross-fertilization. We roughly 1000 eggs from each colony to 1 l of filtered seawater and 10 ml of sperm suspension ($1–7 \times 10^7$ sperm/ml) from one colony to egg suspensions. For Experiments 3 and 4, 5 days after the fertilization during the larval stage, we divided the larvae from each treatment (S: thermal stress fragments and C: control fragments) into two groups. Larvae from control treatments were divided into CC (control condition) and CS (stress condition). Larvae from stress treatment were divided into SC (control condition) and SS (stress condition) for larvae and recruits thermal stress experiments (Fig. 1). The thermal stress experiments on larvae were followed by measuring larval settlement rates and recruits under thermal stress ($31.2 \pm 0.4$ and $31.1 \pm 0.5$ °C ($\pm$SD) and control $26 \pm 0.4$ and $28.4 \pm 0.5$ °C ($\pm$SD) conditions, respectively. The control conditions were ambient temperature in early summer. The water temperatures changed within a day and during experimental days. The temperature in the thermal stress treatment was increased at a rate of ~0.5 °C every 6 h and then maintained for 21 days. The water temperature was monitored every 10 min and recorded using a temperature logger (HOBO Water Temp Pro v2, Onset Computer Corporation, Bourne, MA, USA) throughout the experiment. Each treatment included five replicates, and all tanks were filled with seawater and equipped with a heater.

### Carryover effects of thermal stress on oogenesis and subsequent larvae

*Fecundity (number of eggs per bundle).* In the present study, we counted the number of eggs per bundle. Initially, after the bundles floated to the surface during the spawning, twenty replicates were gently and randomly taken with pipettes from each control (CF) and stressed fragment (SF). Then, after the bundles were become apart (usually within half an hour of the release), the number of eggs was counted under a stereomicroscope. We measured six bundles from each fragment from four spawned stressed fragments and six control fragments.

*Volume of eggs.* Twenty eggs were gently and randomly taken from each treatment when the bundles were fallen apart. Then, the samples were photographed using a microscope (Olympus BX53, Olympus Corporation, Tokyo, Japan). We measured the longest and shortest axis (length and width, respectively) of each egg using Image J assuming[67] an elliptical shape. Egg volume (EV) was calculated using the formula $EV = 4/3 \ \pi ab^2$, where $a = $ ½ egg length and $b = $ ½ egg width.

*Size of the larvae.* At 5 days post-fertilization (dpf), we randomly collected 50 larvae from each treatment and photographed them under a microscope (Olympus BX53, Olympus Corporation, Tokyo, Japan) using a CCD camera (Olympus DP 72) at 10× magnification. The larval length was measured from the images using ImageJ software.

*The survival rate of the larvae under thermal stress.* In each treatment, one hundred 5 days post-fertilization (dpf) larvae were randomly selected and transferred into five glass replicates containers in each temperature treatment. We set two tem-perature treatments (1) C: control, $26 \pm 0.4$ °C ($\pm$SD); (2) S: thermal stress, $31.2 \pm 0.4$ °C ($\pm$SD) (five replicates from control or stressed fragments × two tem-perature treatments). We filled the containers with 300 ml of seawater (replaced every day). For survival analysis, we counted the number of live larvae in each treatment daily for 21 days.

*Lipid extraction and calculation.* We measured the amount of lipids in the larvae according to a previously described method[55]. From each treatment (control and stress fragments), five replicates of 50 swimming larvae were randomly selected from larval stocks on the first and after 21 days of the experiment to obtain the total amount of lipid in larvae. First, we performed lipid extraction using intensive homogenization in a 6:4 mixture of dichloromethane/methanol. The mixture was evaporated using an Eyela NVC-2100 rotary evaporator (Japan), washed using deionized water to remove other materials, and centrifuged at $2500 \times g$ for 10 min. Then, the extracts were evaporated, and the removed remnant water with nitrogen gas passed through a short bed of $Na_2SO_4$ into a pre-weighed 4-ml glass vial (Supelco, pre-burned at 400 °C for four h). Finally, the vials with lipid content were weighed to the nearest 0.1 µg using a microbalance to estimate the lipid's dry weight per larva.

*Settlement behavior and size.* We compared the settlement rates of 5 days post-fertilization (5dpf) larvae in each condition (CC, CS, SC, and SS). We put pre-conditioned limestone tiles in small containers filled with filtered seawater for each treatment. Five replicates were conducted in each condition. One hundred larvae were introduced to each replicate in the container to investigate the settlement rate. We counted metamorphosed larvae after 24 and 48 h with a stereomicroscope (Olympus SZ61). In addition, we photographed the settlements with a digital camera (Canon G10) (Supplementary Fig. 1d) and measured the highest diameter of twenty recruits using ImageJ software (version 1.51).

*The growth rate of recruits.* To test whether thermal stress affects the growth rate of recruits, we compared the growth of recruits in each of the four treatments (CC, CS, SC, and SS) over two months from the first day of settled larvae to two months of recruits. After 21 days of thermal stress experiments on recruits, the recruits were maintained at ambient temperature in outdoor tanks with flow-through seawater under natural solar irradiation. After two months, recruits from each treatment were photographed (Supplementary Fig. 1e), and the longest diameter (mm) of twenty recruits was randomly measured from images of recruits using ImageJ software.

*Survival of recruits under the thermal stress.* To evaluate the survival of recruits in each condition treatment (CC, CS, SC, and SS), we maintained recruits in different treatments (thermal and control conditions) for 21 days. The recruits were counted with a stereomicroscope (Olympus SZ61). The seawater was changed daily.

**Statistics and reproducibility.** Before conducting statistical analysis, we examined the normality and homogeneity of the data using the Shapiro–Wilk and Levene's tests, respectively. To assess the differences in brightness and *Fv/Fm* among adult colonies over the experimental period, we used an LMM. The models were fitted using the 'nlme' package in R statistical software and selected according to Akaike's Information Criterion[68]. We examined the symbiotic algal density and chlorophyll content of adult colonies among treatments using an unpaired *t*-test. In addition, we applied the Kaplan–Meier log-rank test to analyze survivorship in adult colonies, their larvae, and the recruits in different treatment groups. To compare the number of eggs per bundle, egg volume, and larval size between groups, we used an unpaired two-sampled Wilcoxon test. Larval settlement rates between groups were compared using a non-parametric Kruskal–Wallis test or pairwise Wilcoxon test with multiple comparisons. We used a linear model (LM) to compare lipid con-tents and larval settlement size between treatments. All statistical analyses were performed using R statistical software (v.4.0.4)[68].

**Reporting summary.** Further information on research design is available in the Nature Portfolio Reporting Summary linked to this article.

### Data availability
All data supporting the findings of this study are available within the article and its Supplementary Information files (Supplementary Data 1). Additional information, and relevant data will be available from the corresponding author upon reasonable request.

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

## Acknowledgements

We would like to thank Dr. S. Harii and F. Sinniger for useful comments and the staff of the Sesoko Station for their support. This work was partly supported by the Project Foundation of the Tropical Biosphere Research Center of the University of the Ryukyus and JSPS KAKENHI (21H05304, 22H02369 to MM).

## Author contributions

S.H.-K. designed research; S.H.-K. performed research; S.H.-K., and M.M., P.T.-K. analyzed data; S.H.-K., M.M., T.N., P.T.-K., S.K. contributed new reagents/analytic tools; S.H.-K., and M.M. wrote the paper. All authors commented and approved the manuscript.

## Competing interests

The authors declare no competing interests.

## Ethics

The methods were performed in accordance with relevant guidelines and regulations and approved by University of the Ryukyus.
