## [Peer Review File · Communications Biology]

Reviewers' comments:

Reviewer #1 (Remarks to the Author):

The paper examined the carryover effects of thermal stress on adult fecundity and fitness of early life stages of the reef-building coral *Acropora tenuis*. They counted the number, size, and lipid content of eggs from colonies that they had previously exposed to control or stress temperatures. They then investigated whether the larvae obtained from control or stressed parents will exhibit tolerance to the same conditions. Their findings revealed that colonies subjected to stress produced smaller eggs but in larger numbers. However, larvae and juveniles appeared to perform similarly in terms of survival and settlement at elevated temperature. Marginal improvement in survival was reported for juveniles from stressed corals, though the numbers were not robust.

The materials and methods section is lacking details important for evaluation of experimental design and for reproducibility of the study. The results are concisely described and figures are easy to understand. The discussion is very limited, with most parts just reiterating what was already stated in the results. This section could be further enhanced by relating or comparing the results to other studies that have conducted similar work looking at impacts of parental thermal exposure on larval tolerance (e.g. Mcrae et al., 2021; Dixon et al., 2015; Howells et al., 2021), as well as discussing coral strategies for responding to stress or improving fitness. Nevertheless, the study is interesting and relevant to our understanding of coral reproduction and larval biology amidst global warming.

Specific comments:

Title

Line 1: Suggest to indicate the coral species name in the title

Keywords

Line 16-17: Avoid using keywords that are already mentioned in the title (e.g. coral, fitness)

Abstract

Line 23: Italicize "*Acropora tenuis*"

Line 28: The results show marginally higher rates of survival of juveniles from parents that experienced bleaching. Please revise this sentence.

Line 29-30: This line can be omitted because the influence of symbiotic relationships on larval survival was not examined in the study.

Line 30: What are these adaptive strategies? Please specify.

Introduction

Line 54-56: Omit "Given the widespread degradation of coral reefs"

Line 56: This statement seems out of place.

Line 57-58: Unclear what is meant by this statement. Please rephrase.

Line 62: Elaborate on these "previous studies"

Line 64: This sentence can be added to the last paragraph. Better if the hypothesis or expected results are mentioned at the end of the introduction.

Results

Line 69: A little bit of introductory sentence should be added at the start of the results section because methods appear at the end of the manuscript. It would be useful to include as supplement some photos attesting to the bleaching response of the coral fragments.

Lines 75-82: Provide the values for algal density and chlorophyll a and c.

Line 81: Where is it shown that chlorophyll c is significantly different? Fig1E and F say no significant differences in chl a or chl c content. Fig1E and F should be cited in this section.

Line 83-84: From the plot, it seems like survivorship after 10 months is lower than 83%. Fig1C is

mentioned after Fig1F. Figure panels should be cited in consecutive order.

Line 85: How many corals survived up to spawning in 2019? How many spawned and how many were used for fertilization to obtain gametes used in Experiments 2-4?

Line 95: "SF larvae exhibited substantial thermal tolerance" may be misleading as based on the data these performed similarly to CF larvae under the treatments. Fig3E is cited before Fig3C and D.

Line 102: Supplementary Table 1?

Line 116: It is interesting to note that settlement was faster at elevated temperature for larvae from either parent type.

Line 132: Supplementary Table 2?

Discussion

Line 141-142: What is the significance of having more eggs despite smaller size?

Line 156-157: Why might the rate of lipid utilization in SF be lower than in CF larvae and what would the implications of this be?

Line 158: it is not true that the SF larvae were not vulnerable to stress

Line 159: both CF and SF larvae settled rapidly under elevated temperature

Line 163-165: Did the larvae in the experiment remain aposymbiotic throughout?

Line 168-169: Why is having smaller polyps an undesirable characteristic of corals. Please explain further or provide literature supporting this claim.

Line 174: Not sure that it's correct to say that the corals "developed tactics to increase fitness" – perhaps better to simply state what the adaptive strategies were that you observed

Materials and Methods

Line 185: Better to separate adult thermal stress experiments from (lines 186-201) from spawning and larval experiments (lines 202-217).

Line 190: When were fragments collected for initial experiments? Please provide photos of the fragments and the set up with fragments used in this study.

Line 194-195: What is meant "visible bleaching" and was this only observed on Day 10? Please indicate.

Line 202: When was spring 2019?

Line 204: How many corals from each treatment were collected?

Line 205: Were all colonies kept in separate tanks for gamete collection? How did you ensure no cross-fertilization between colonies?

Line 209: Further details needed on number of parents from which gametes were obtained for fertilization, amount of eggs/sperm mixed, and fertilization rates

Line 210-211: Experiments-> Experiments

Line 218: Section should appear after lines 186-201

Line 225: The four classifications of larvae (CC, CS, SC, SS) were not explained and introduced in the methodology part. You can mention those here.

Line 226: How was gamete count done? How many bundles were counted? How many eggs were measured for size? How many eggs were used for lipid analysis?

Line 230: Even though the method on how to measure the lipid content of the larvae is based on a paper, please provide a brief description of this method here.

Lines 233-236: Please provide relevant details on the larval experiments (how old were the larvae, what larval density, how was temperature manipulated, etc).

Line 236-237: Was this for larval size measurement? Please indicate

Line 240-243: Please provide details on the settlement experiment. What substrate was used, what was larval density, how were settled larvae counted, how was water quality maintained over 21 days?

Line 243: spats > settled juvenile corals

Line 246: cells > recruits?

Figures

Line 419: "not significantly"

Figure 3D: Instead of showing "first" and "last" on x-axis, specify the days (Day #).

Figure panels should be cited consecutively in the text.

Reviewer #2 (Remarks to the Author):

In this study by Hazraty-Kari et al., they report how a multi-year thermal stress experiment can impact reproductive output and quality of offspring from a broadcast spawning coral species. From this work, they found that heat stress increases egg number but lowers lipid content, suggesting trade-offs in gametes from the stress event. However, there was no evidence of stress hardening in the larval survival from parent colonies exposed to thermal stress. *A. tenuis* larvae and recruits already has a fairly high survival rate to heat stress compared to other species, and therefore may be more resilient to thermal events. Hence, why they are good at re-seeding the reef post mortality events.

To me, the paper needs to bolster their motivation for the study more. The experimental design is setup to address heritability or maternal effects as a form adaptation or acclimation in offspring to thermal stress of the parents. However, the authors do not touch on the large body of literature on this topic of "stress-hardening" offspring. This work is interesting because there was no evidence of improved performance of the offspring for the stressed fragments, but other differences were observed. Focusing the introduction and discussion in light of the prior papers using comparable experimental designs would enhance the future citation of this work.

Are all offspring products of self-fertilization? The breeding scheme was not clear to me. I interpreted that all the original fragments were from one colony. Is that correct? Perhaps I missed the description of this in the methods or supplement.

I believe the author's need to revise the manuscript prior to publication.

Other comments:

Line 22: I would reconsider the possessive use of stress's in this sentence. An alternative might be, "We examined the direct and carryover effects of thermal stress on fecundity..."

Line 23: *Acropora tenuis* should be italicized.

Line 28: I recommend removing "potentially".

Line 29: As far as I can tell, symbionts were not introduced to the larvae/recruits in this study. While I agree with this statement, I do not think it should be featured so prominently in the abstract without additional context. Rather highlight the lipid differences between the treatments first and then mention how symbionts may contribute or not to larval survival. In fact, most of the results are absent from the abstract after reading the rest of the paper.

Line 57: I believe the authors mean, the larvae were released by adult colonies and then settled during high-temperature conditions?

Line 64: This floating sentence should perhaps be combined with the previous paragraph or developed further.

Line 86-87: I believe additional clarification could help the reader here that the number of eggs increased for the SF replicates that did spawn.

Line 106: This is the first mention of planula. I would recommend either sticking with larva(e) throughout the paper or define this word that is may be uncommon in other fields.

Line 235: How many larvae were used for the survival analysis? Refer to supplemental data here.

Line 232: How was lipid content measured?

Line 246: Cells were counted? What cells?

Reviewer #3 (Remarks to the Author):

Dear authors, dear editor,

This study entitled "Adaptive strategies to increase fitness after thermal stress in the reef-building coral *Acropora*" by Hazraty-Kary et al. investigates the cross-generational acclimation potential of different life stages of the tropical coral *Acropora tenuis* to thermal stress in several short-term aquarium experiments. Adult coral fragments were exposed to a heat stress treatment and the effect on the next generation (eggs, larvae and juvenile corals) was studied in different temperature treatments in the following year. This study claims that the heat stress treatment of the adult coral fragments improves the heat stress tolerance of the next generation.

This manuscript addresses a novel topic by investigating the effect of elevated temperature on different early life stages of corals and how temperature increases affect the next generation with a novel experimental design. The manuscript is well written and the data in the graphs are presented very well and easy to understand. The visualisation of the experimental design in Fig. 1 is very helpful for the reader. However, this manuscript is missing important information to clearly understand how the experiments were conducted and to be able to reproduce them. Partly, this information is provided in the supplementary material (but should be moved to the main text) and partly, more methodological information has to be provided (see specific comments below). In addition, this manuscript is missing detailed information from the literature in both the introduction and especially in the discussion, where the findings are not appropriately discussed. Therefore, some more background information should be provided in the introduction (incl. detailed information about the objectives, the experimental design and the research question(s) of this study) and the findings should be discussed in much more detail and in a broader context of the available literature (see detailed comments below). The discussion should rather focus on possible explanations for the main findings of this study in the context of the literature and what this means for the reproductive process of corals in the future under increasing water temperatures instead of mainly repeating the main results. With these fundamental changes in the introduction, the methods section and the discussion, I think this study will be very interesting for the audience of this journal.

Comments:

1. Please ensure consistent use of terminology in the whole manuscript and the figures as this will make it easier for the readers to understand what you are talking about. For example, you sometimes write "recruit" and sometimes "juvenile", "control" and "normal" treatment, "oocytes" and "eggs", "bleaching" and "brightness", "larvae settlement" and "larvae settling".

Title

1. I would suggest to already state in the title that this study was conducted with *Acropora tenuis* and that it focusses on different life stages of this coral species. As this is the main focus of the manuscript and a novel research topic, I would definitely highlight this already in the title, e.g. "Adaptive strategies of different life stages of the coral *Acropora tenuis* to elevated temperature".
2. I suggest to delete the key words "coral" and "*Acropora tenuis*" as they are already mentioned in the title (or should be in my opinion as suggested above).

Abstract

3. Line 22: What do you mean by "strategies to survive for the next generation"? Please specify.
4. Line 23: *Acropora tenuis* in italics. Please also specify which parameters you have measured in order to be able to make statements about the fitness of the corals.
5. Line 24: Please include the duration of the experiment.
6. Line 26: State which parameters were measured for the juvenile corals. As you only investigated them for 21 days, I would suggest to rather call them coral recruits instead of juveniles (as you also sometimes do in the manuscript).
7. Line 27: It is not clear why you explicitly state that you only investigated the "stressed corals" as the control corals were also investigated I guess, please specify.
8. Line 28: You do not state any results that support your statement of "increased larval production" as you only examined the "egg production". Is this what you mean here? Please clarify.
9. Lines 28ff: The statements of the main results are not very clear in the abstract: e.g. "potentially higher rates of survival for larvae and juveniles", but you found highest survival rates of recruits in control treatments (lines 129ff). Or do you mean the higher survival rate in SS compared to CS here (line 133)? Which results support your statement of "Symbiotic relationships may assist the survival of larval stages with lower lipid content"? Please state the results more clearly in the abstract.

Introduction

10. As you investigate the effect of elevated temperature on adult corals and early life stages, please provide some more general background information in the introduction about what is known so far about the effect of increasing temperature on different life stages (larvae, recruits and adults) of corals to make more clear what is novel about this study. I include some references that I think might be useful to include in the introduction and which can also be interesting for the discussion. Even though the study by Putnam & Gates (2015) investigates the trans-generational acclimatisation of corals under elevated temperature and pCO_2 , I think it is still important to include this study in this manuscript to show what has already been done in previous similar studies.

Anlauf, H., D'Croz, L., & O'Dea, A. (2011). A corrosive concoction: The combined effects of ocean warming and acidification on the early growth of a stony coral are multiplicative. *Journal of Experimental Marine Biology and Ecology*, 397, 13–20.

Bahr, K. D., Tran, T., Jury, C. P., & Toonen, R. J. (2020). Abundance, size, and survival of recruits of the reef coral *Pocillopora acuta* under ocean warming and acidification. *PLoS ONE*, 15, e0228168.

Chua, C. M., Leggat, W., Moya, A., & Baird, A. H. (2013). Temperature affects the early life history stages of corals more than near future ocean acidification. *Marine Ecology Progress Series*, 475, 85–92.

Foster, T., Gilmour, J. P., Chua, C. M., Falter, J. L., & McCulloch, M. T. (2015). Effect of ocean warming and acidification on the early life stages of subtropical *Acropora spicifera*. *Coral Reefs*, 34, 1217–1226.

Jiang, L., Zhang, F., Guo, M. L. et al. (2018). Increased temperature mitigates the effects of ocean acidification on the calcification of juvenile *Pocillopora damicornis*, but at a cost. *Coral Reefs*, 37, 71–79.

Liberman, R., Fine, M., & Benayahu, Y. (2021). Simulated climate change scenarios impact the reproduction and early life stages of a soft coral. *Marine Environmental Research*, 163, 105215.

Negri, A. P., Marshall, P. A., & Heyward, A. J. (2007). Differing effects of thermal stress on coral fertilization and early embryogenesis in four Indo Pacific species. *Coral Reefs*, 26, 759–763.

Putnam, H. M., & Gates, R. D. (2015). Preconditioning in the reef-building coral *Pocillopora damicornis* and the potential for trans-generational acclimatization in coral larvae under future climate change conditions. *The Journal of Experimental Biology*, 218, 2365–2372.

Randall, C. J., & Szmant, A. M. (2009a). Elevated Temperature Affects Development, Survivorship, and Settlement of the Elkhorn Coral, *Acropora palmata* (Lamarck 1816). *Biological Bulletin*, 217, 269–282.

Randall, C. J., & Szmant, A. M. (2009b). Elevated temperature reduces survivorship and settlement of the larvae of the Caribbean scleractinian coral, *Favia fragum* (Esper). *Coral Reefs*, 28, 537–545.

Ross, C., Ritson-Williams, R., Olsen, K. et al. (2013). Short-term and latent post-settlement effects associated with elevated temperature and oxidative stress on larvae from the coral *Porites astreoides*.

Coral Reefs 32, 71–79.

Schnitzler, C.E., Hollingsworth, L.L., Krupp, D.A. et al. (2012). Elevated temperature impairs onset of symbiosis and reduces survivorship in larvae of the Hawaiian coral, *Fungia scutaria*. *Mar Biol* 159, 633–642.

11. As this manuscript investigates different coral life stages, please provide some information about the life cycle of *A. tenuis*.
12. Line 42f: I think it would be helpful for the reader to provide more information from the literature about the different mechanisms how corals can acclimate to elevated temperatures.
13. Line 43: Can you include examples of more heat-tolerant species?
14. Line 45: "Physiological performance" is a very general term. What do you mean exactly? Could you please specify and also provide some more recent references here?
15. Line 48: I would delete "young corals" as you also write "early life stages".
16. Line 55f: You state that "the thermal stress history of parents can influence the thermotolerance of early life stages in corals and other marine organisms (30, 31)", but ref. 31 is an acidification experiment. Please provide another suitable reference here.
17. Line 57f: I do not understand this sentence "Thus it is highly likely that..., resulting in high-temperature conditions" as not the coral larvae lead to elevated temperature, please rephrase.
18. Line 58f: Can you provide more information about how much the temperature fluctuates in the habitat of *A. tenuis*? As many tropical coral species are experiencing high temperature fluctuations, is this species subject to especially high temperature variability?
19. Line 60f: Please specify how responses to elevated temperature vary across species.
20. Line 61f: References missing for "...the effects of thermal stress on adults, larvae, and juveniles have been reported in previous studies".
21. Line 64f: Please provide more information about the objectives of this study in the last paragraph of the introduction as this is not clear yet, incl. a short statement of the research questions/objectives you tried to answer with this experiment, a short summary how the experiments were conducted (experimental design, referring to Fig. 1) and which parameters were measured. I would also suggest to state here that this study was conducted with *A. tenuis*.

Methods

22. In my opinion, the methods section is missing some important information to fully understand how the experiments were conducted. I would suggest to move all the information that is already provided in the supplements, but is essential to fully understand the experimental design, to the main text.
23. Line 185: I suggest to shorten subheading to "coral collection and experimental design".
24. Line 186: When was the experiment conducted?
25. Line 187: I suggest to replace "adults" by "adult physiology" (or more precise term) as you also state that you measured "adult fecundity" to make the difference between the two more clear.
26. Line 189: How many fragments were used? How many replicates in how many separate aquaria?
27. Line 192: How was ensured that the temperature in the control treatment was stable during the experiment when a heater was only placed in the heat stress treatment? What was the water flow in the flow-through system? How were constant conditions ensured with the constant flow of water?
28. Could you please provide photographs of the aquarium system in the supplements to better understand the setup of the aquarium system?
29. Line 193: Please state the exact mean +/- standard deviation for both temperature treatments. Also mention how and how often the temperature was measured during the experiment. Please explain why you decided to conduct a +3 °C heat stress treatment. Why did you decide to conduct the experiment at 31 °C and not at a higher or lower temperature? Was this decision based on the temperature range that the corals experience *in situ* or that they will experience until the end of this century? Then please provide evidence for this.
30. Did you also measure other seawater parameters such as salinity, oxygen concentration and pH during the experiment and could you please also provide this information (mean +/- SD)?
31. Line 184f: What exactly do you mean by "until the fragments started to show visible bleaching"? Please provide some more information here.
32. Line 195f: How was the 12-h light:12-h dark cycle controlled? Were the corals only kept in the

outdoor tank for the acclimation period as mentioned in the supplements or also during the experiments?

33. Line 197f: As you regularly took pictures of the corals, I think it would be interesting for the readers to see pictures of these corals after the heat stress experiment to see the degree of bleaching (see my comment above). Please provide some pictures of the corals in both treatments as examples.

34. Line 199f: "Physiological parameters (including the symbiont algal density and chlorophyll content)": Which other measurements were conducted if you write "including"? When exactly were the measurements conducted? How many days before/after the experiments? Do you also have data about the growth or respiration rates of the corals over the time of the heat stress treatment and under control conditions?

35. Line 200f: Please state why you did not bleach the corals right before the spawning event but decided to bleach them one year before the spawning. This is not clear but important to understand the reasoning for this part of the experimental design.

36. Line 202f: When exactly were the corals re-collected in spring 2019, when were the experiments conducted and when did the spawning event occur?

37. Line 206f: Here you state that you used two treatment conditions after re-collection of the corals but experiment 2 in 2019 seems to have been only conducted under control conditions in Fig. 1 (blue aquaria), please clarify.

38. Line 208: How many aquaria and how many fragments in each aquarium were used?

39. Line 218: When were the physiological parameters of the adult corals measured? I assume in 2018. Therefore, I would write this paragraph before you describe the experiments in 2019 in order to describe the experiments in chronological order (and the same order as in Fig. 1), which will make it easier for the readers to follow.

40. Line 232: When were the larvae for the lipid analysis sampled and how were they processed? I assume also according to ref. 44?

41. Line 234ff: Please describe the setup of the glass containers in more detail: How were the five glass containers set up? How was water movement inside the containers ensured? How was the water replaced without losing any larvae? How many larvae were in each glass container? How did you measure the water temperature inside the containers and how did you ensure stable temperatures? In Fig. 1 it looks like the glass containers were placed into aquaria, were the five containers placed in the same aquarium or in separate ones? Where did the larvae settle on? In Fig. 1 it looks like the larvae settled on small plates that were placed in the aquaria afterwards, is this correct? Please provide more information here. You write "five replicates x two treatments" but according to Fig. 1 you had four treatments (CC, CS, SC, SS), please clarify. Please also state in the text what the abbreviations CC, CS, SC, SS stand for as this is only partly explained in the figure legends.

42. Line 237ff: Please state why you took pictures of the larvae and also provide example pictures of the larvae of the different treatments if you observed any differences.

43. Line 240ff: Please state that the settlement rate was investigated over 2 days. How was the size of the settled planulae/recruits measured, what did you measure exactly?

44. Line 246: It is not clear what you mean by "the cells were counted", do you mean the number of early recruits? Could you also provide pictures of the young recruits for the different treatments?

45. Line 258: Which distribution of the data was used for the generalized linear model, i.e. which "family" in the glm function?

Results

46. Line 69f: What is meant with "brightness"? How is this measured? As you apparently mean "bleaching" here, I would change the term in the text to make this more clear. As you observed significant differences between the treatments, please also provide some pictures of the coral fragments.

47. Line 70f: I would suggest to delete the sentence "We also measured..." as this was already described in the methods section.

48. Line 76: I think it is a bit confusing to talk about "highest" here as you only have two treatments, please rephrase.

49. Line 79f: Do you mean Chl a here?

50. Line 81f: You state here that the chlorophyll c₂ content was significantly different but this is not shown in Figure 2F, please correct.
51. Line 83f: Please add here the information over which time period the survival was measured.
52. Line 86f: The sentences "One SF did not spawn at all during the spawning period. However,..." are not clear as you talk about just one fragment in the first sentence but about all fragments in the second sentence, I assume. I would suggest to rephrase these sentences and delete "however".
53. Line 95: "SF larvae exhibited substantial thermal tolerance": How does this statement fit with the results as the larvae in both thermal stress treatments had lowest survival rates.
54. Line 96f: The survival rates of the treatments CS and SS are exactly the same, is this correct?
55. Line 102: I guess you mean "Supplementary Table 1" here?
56. Line 103: The sentence "To estimate larval size,..." should be stated in the methods section and can be deleted here.
57. Please provide a table with all the results of the statistic tests (not just the p-values) in the supplements, not only for the results of the survival analyses.
58. Line 121f: The sentence "In each treatment,..." should be stated in the methods section and can be deleted here.
59. Line 123ff: Here, it is stated that the CC recruits are significantly larger than CS, SC and SS, but this is not shown in the figure. Please also include the statistical results in the figures and do not only show the significant results of CC-SC.

Figures

60. I would suggest to use the same four colours in Fig. 1 for experiments 3 and 4 as in figures 3 and 4 (incl. light blue for CS and pink for SS) and dark blue for CF for experiment 1 and 2 to make it more clear for the readers. I would also use the four colours in Fig. 4B instead of just red and blue.
61. Line 401f: The control group was also maintained in the aquarium for 10 days, correct? Please clarify.
62. Line 403: I would suggest to write "larvae settlement" instead of "larvae settling" (also in the figure) as in other papers and other parts of the manuscript.
63. Line 404f: But the juvenile corals of the control treatments were also investigated, but this is not stated here, please clarify.
64. Line 407f: What is meant with "brightness"? How is this measured? As you apparently mean "bleaching" here, I would change the Y-axis label and the term in the figure legend and in the text. I would suggest to write "survival" instead of "mortality" as this is also written in the figure. As you also plot the symbiont density, Chl a and Chl c₂ here but this is not mentioned in the figure legends, maybe change the figure legend to "Effect of thermal stress on the physiological performance and survival of adult coral fragments (Experiment 1)" or something similar. I would also mention somewhere in the figure legend that these data are from *Acropora tenuis* as figure legends should be understandable without reading the main text.
65. Line 409f: How did you determine the Y-axis values of 50 and 225 for healthy and bleaching? And what is the unit for the Y-axis?
66. Line 413ff: I do not understand the survival graph of Fig. 2C. According to the figure legend you had 6 fragments per treatment, but according to the graph 8 fragments died over the 10 months but this only represented about 25 % of the fragments.
67. Figure 3: Please specify in the figure legend how many fragments per treatment were used to count the number of eggs in the bundles in Fig. 3A. It sounds as if n = 40 fragments were used in Fig. 3B but if I understood it correctly, these are the number of egg bundles, correct? Please clarify. Why were 30 bundles used for Fig. 3A but 40 bundles for Fig. 3B? What does "first" and "end" on the X-axis of Fig. 3D mean? Please specify in the figure and the figure legend as this is not clear when the lipid concentration was measured. On the X-axis of Fig. 3E are two times 0 days instead of 0 and 5 days. In Figure 3E, I suppose you had four treatment groups in the end. I think it is a bit confusing to talk about "two treatments" here.
68. It is not clear what the settlement size in Fig. 4B is. Is this the diameter or height of the coral recruit or something else? How was this measured? Please specify in the methods section and in the figure legend.

69. Do you have any information about the growth rates of the coral recruits during the 21 days after settlement? As you took pictures of the coral recruits, I would suggest to also measure their growth and include this information in the manuscript as this is a very important trait in my opinion. Even if most recruits were able to survive the first 21 days after settlement in the four treatments, there might be differences in growth rates that are important to also take into account for the development of the coral recruits.

Discussion

70. In my opinion, a thorough revision of the discussion and conclusion is necessary and I will therefore only comment on general things and not on details at this point. Please focus less on the results of your experiments (which are already stated in the results section) and more on (potential) explanations for your findings, do not just repeat the results here as you do in large parts of the discussion. Please discuss your findings in much more detail instead. In cases where no other studies are available yet, I would like to see some speculation about your findings and what you think this means for the corals. Please also discuss all findings of this manuscript (all measured traits) and if you think that the findings of some of the traits are not important enough to be discussed, then please do not show the results in the manuscript or move them to the supplements (e.g. the physiological data of the adult coral fragments are not discussed at all, whereas a figure showing the lipid content depending on egg size is missing).

71. Please discuss that most results of the larvae and recruits do not seem to be related to the heat stress treatment of the adult corals (Experiment 1) but rather to the heat stress treatments in Experiments 3 and 4. I think this is a very important point and should be discussed (also see comment below to Fig. 4E).

72. As you mention in the introduction, marine heat waves will increase in the future and corals will experience warmer temperatures all over the world. I think it is therefore very important to also discuss what the results of your study mean for the reproductive success of adult corals and the development of larvae and recruits in the future in a warmer ocean in more detail.

73. Line 149: Where are the results shown that "the lipid content of coral eggs differed with egg size?" I did not find any figure that shows this or any information about this in the results section.

74. Line 158: But the survival rate of larvae seems to depend more on the heat stress treatment in experiment 3 (as shown in Fig. 3E) than the heat stress treatment of the adult coral fragments. How do you come to the conclusion that the larvae of the stressed coral fragments were not vulnerable to thermal stress?

75. Line 160: Same as previous comment: How do you come to the conclusion that the SF juveniles were more tolerant to high water temperatures? I think if you discuss this here that SS were slightly more tolerant than CS, then you should also discuss that they were nevertheless both less tolerant than CC and SC. The response of larvae and juveniles seems to depend more on the heat stress treatment in experiments 2-4 than in experiment 1.

76. Line 176ff: In my opinion the strong statement "Corals also exhibited thermal acclimation under thermal stress. The acclimation potential of a subsequent generation identified in this study indicates the possibility that future generations may survive conditions of thermal stress" is not supported by the results of this study as survival rates only slightly increased in SS compared to CS, please rephrase and generalise less.

Supplementary information

77. "Four different life stages of corals: adults, fecundity and eggs, larval settlement, and early recruits." However, "fecundity" and "larval settlement" are not life stages of the corals. I would suggest to either only state the life stages that you investigated or the parameters that were measured.

78. "...until the fragments started to show a visual bleaching response" and "The measurements were repeated every day until the signs of bleaching were revealed approximately ten days after the thermal stress": Was the experiment conducted for exactly 10 days as stated in the main text or did the experimental duration differ for each fragment? Or did it just differ when the fragments started to bleach? In that case, when did you decide to stop the experiment? Were all fragments bleached after

10 days? This is not clear, please clarify.

79. It is not really clear how the surface area of the corals was measured, please rephrase this sentence. I assume that the surface area was determined based on the paraffin wax dipping technique. How did you determine the "mean surface area" for each fragment? Did you repeat this procedure several times as you write that only one branch of each fragment was used for this analysis?

80. Why was only one branch per fragment used to determine the algal symbiont densities and chlorophyll a and c_2 content? Do you have any information about the variability of these parameters in the same coral fragment or in *A. tenuis* in general?

Reviewers' comments:

Reviewer #1 (Remarks to the Author):

The paper examined the carryover effects of thermal stress on adult fecundity and fitness of early life stages of the reef-building coral *Acropora tenuis*. They counted the number, size, and lipid content of eggs from colonies that they had previously exposed to control or stress temperatures. They then investigated whether the larvae obtained from control or stressed parents will exhibit tolerance to the same conditions. Their findings revealed that colonies subjected to stress produced smaller eggs but in larger numbers. However, larvae and juveniles appeared to perform similarly in terms of survival and settlement at elevated temperature. Marginal improvement in survival was reported for juveniles from stressed corals, though the numbers were not robust.

The materials and methods section is lacking details important for evaluation of experimental design and for reproducibility of the study. The results are concisely described and figures are easy to understand. The discussion is very limited, with most parts just reiterating what was already stated in the results. This section could be further enhanced by relating or comparing the results to other studies that have conducted similar work looking at impacts of parental thermal exposure on larval tolerance (e.g. Mcrae et al., 2021; Dixon et al., 2015; Howells et al., 2021), as well as discussing coral strategies for responding to stress or improving fitness. Nevertheless, the study is interesting and relevant to our understanding of coral reproduction and larval biology amidst global warming.

Specific comments:

Title

Line 1: Suggest to indicate the coral species name in the title

Answer: We changed the title and added the species name *Acropora tenuis*.

Keywords

Line 16–17: Avoid using keywords that are already mentioned in the title (e.g. coral, fitness)

Answer: We changed key words as follows; Alternative tactics, Consecutive thermal stress, coral, fecundity, Recruits (Line 17)

Abstract

Line 23: Italicize “*Acropora tenuis*”

Answer: We changed as you suggested (Line 23).

Line 28: The results show marginally higher rates of survival of juveniles from parents that experienced bleaching. Please revise this sentence.

Answer: Thank you for your advice. We changed this sentence to show more clear. “As a result of the thermal stress, we observed a trade-off in egg production; an increase in egg number but a decrease in a size. In addition, larvae from the stressed colony showed marginally higher survival rates.” (Line 26–28)

Line 29–30: This line can be omitted because the influence of symbiotic relationships on larval survival was not examined in the study.

Answer: We omitted this sentence.

Line 30: What are these adaptive strategies? Please specify.

Answer: Here we mean the stressed colony produces more but smaller eggs and heat-tolerance larvae. We changed this sentence to convey our message as follows; “Collectively, heat-vulnerable corals have strategies to produce more heat-tolerant larvae after they experience stress.”(Line 29–30)

Introduction

Answer: According to the other reviewers’ comments, we extensively edited the introduction.

Line 54–56: Omit “Given the widespread degradation of coral reefs”

Answer: We omitted this sentence.

Line 56: This statement seems out of place.

Answer: We omitted this sentence.

Line 57–58: Unclear what is meant by this statement. Please rephrase.

Answer: Thank you for your suggestion. We changed this paragraph to show more precisely about our hypothesis; The coral species *Acropora tenuis* inhabits shallow reefs with temperature fluctuations, and these fluctuating temperatures may affect both larval settlement and juveniles³⁹. Responses to temperature (such as larval recruitment) also vary across species⁴⁰, and the effects of thermal stress on adults, larvae, and juveniles have been reported in previous studies. In a higher water temperature, larval development becomes faster^{22,31}. In addition, metamorphosis leads to settlement occurring faster⁴¹ but also shows higher mortality^{22,31,34}. As a result of faster development of the larval stages, they tend to settle near the adult colony⁴². The larval dispersal becomes a smaller distance, and the number of the recruited larvae could increase after the degradation of the reef due to heavy bleaching events. From this aspect, more heat tolerance larvae and recruits are desirable in a higher water temperature and will be a good strategy for their fitness

after colonies experience heat stress. However, the effects of successive thermal stress on adults and the subsequent effects on their fecundity, the larval stage, and the juveniles have not been examined. In this study, we examined the effects of heat stress on the reproduction and early life stages (oocytes, larvae, larval settlement, and early juveniles) of corals (Fig. 1). (Line 77–90)

Line 62: Elaborate on these “previous studies”

Answer: we add several sentences to show the effect of thermal stress on the larvae and adults. Please see above (Line 77–90).

Line 64: This sentence can be added to the last paragraph. Better if the hypothesis or expected results are mentioned at the end of the introduction.

Answer: Thank you for your suggestion. We extensively edited introduction and show our hypothesis in the last paragraph of the introduction and added several sentences to show our hypothesis as shown above (Line 77–90).

Results

Line 69: A little bit of introductory sentence should be added at the start of the results section because methods appear at the end of the manuscript. It would be useful to include as supplement some photos attesting to the bleaching response of the coral fragments.

Answer: In this study, we exposed to thermal stress for 10 days, but beaching did not occur frequently.

Lines 75–82: Provide the values for algal density and chlorophyll a and c.

Answer: the values were added (Line 100~102)

Line 81: Where is it shown that chlorophyll c is significantly different? Fig1E and F say no significant differences in chl a or chl c2 content. Fig1E and F should be cited in this section.

Answer: We show it in Fig. 2d and e for this. (Line 107)

Line 83–84: From the plot, it seems like survivorship after 10 months is lower than 83%. Fig1C is mentioned after Fig1F. Figure panels should be cited in consecutive order.

Answer: Thank you for your advice. We add sentences and Fig. 2f (Line 108–110).

Line 85: How many corals survived up to spawning in 2019? How many spawned and how many were used for fertilization to obtain gametes used in Experiments 2–4?

Answer: Most colonies (five fragments of the stressed and six fragments of the control, a total eleven fragments) survived, and four fragments for stressed (SF) and six control fragments (CF) spawned. We used gametes from all spawned fragments for fertilization for larvae (five colonies in each experiment). We add a description of the spawning in lines 112 to 113.

Line 95: “SF larvae exhibited substantial thermal tolerance” may be misleading as based on the data these performed similarly to CF larvae under the treatments. Fig3E is cited before Fig3C and D.

Answer: We placed larval size in the first section of this paragraph to show Fig, 3C and 3D, and changed “the SF larvae exhibited substantial thermal tolerance” to “Survival rates of the larvae were different between CF and SF treatment” (Line 132)

Line 102: Supplementary Table 1?

Answer: Thank you for your suggestion, We add “supplementary” (Line 139).

Line 116: It is interesting to note that settlement was faster at elevated temperature for larvae from either parent type.

Answer: Thank you for your advice. We changed this sentence as follows; On day 1 of the larval settlement period, larval settlement rates were faster in thermal stress treatments than in control treatments for larvae from both stressed and control fragments. (Line 141-142).

Line 132: Supplementary Table 2?

Answer: Thank you for the suggestion. We add “supplementary” before table2 (Line 157)

Discussion

Answer: We also extensively edited the discussion and show the edited sentences in red charactres.

Line 141–142: What is the significance of having more eggs despite smaller size?

Answer: The energy investment for egg production may be traded-off. We added one paragraph to discuss more about this topic in lines 183–196.

Line 156–157: Why might the rate of lipid utilization in SF be lower than in CF larvae and what would the implications of this be?

Answer: Thank you for your comments. We add lipid consumption rates in line 219–226.

Line 158: it is not true that the SF larvae were not vulnerable to stress

Answer: We weakened the sentence as follows and discussion more about this topic in line 235 to 247

Line 159: both CF and SF larvae settled rapidly under elevated temperature

Answer: We add more discussion about settlement rates in higher water temperatures in lines 227 to 234.

Line 163–165: Did the larvae in the experiment remain aposymbiotic throughout?

Answer: We did not check the symbiotic or aposymbiotic, but we used filtered seawater throughout the experiments, and thus we predicted that most of the larvae were aposymbiotic (Line 213 to 216).

Line 168–169: Why is having smaller polyps an undesirable characteristic of corals. Please explain further or provide literature supporting this claim.

Answer: We add an explanation about this and add the reference in lines 227 to 234.

Line 174: Not sure that it's correct to say that the corals "developed tactics to increase fitness" – perhaps better to simply state what the adaptive strategies were that you observed

Answer: Thank you for your advice. We changed this sentence to simply state as follows; "We also found that colonies that experienced heat stress developed tactics to increase larvae and recruit survival in higher water temperatures" (Line 251–252)

Materials and Methods

Answer: We also extensively edited the method section to show more precisely and moved many parts from supplementary information.

Line 185: Better to separate adult thermal stress experiments from (lines 186–201) from spawning and larval experiments (lines 202–217).

Answer: We separated the adult experiments and larval experiments (adults from 301–383; larvae 384–424)

Line 190: When were fragments collected for initial experiments? Please provide photos of the fragments and the set up with fragments used in this study.

Answer: We add new supplementary information for this supplementary figure 1.

Line 194–195: What is meant “visible bleaching” and was this only observed on Day 10? Please indicate.

Answer: We apologize for not describing precisely, here we did not observe bleaching in the stressed colonies. Then, omit this sentence.

Line 202: When was spring 2019?

Answer: This is June 2019.

Line 204: How many corals from each treatment were collected?

Answer: We collected six colonies and divided into two fragments from each colony (12 fragments). (Line 273–277)

Line 205: Were all colonies kept in separate tanks for gamete collection? How did you ensure no cross-fertilization between colonies?

Answer: We placed each fragment in small brackets before the spawning (As you know, bundles appears on the surface of the colony about 1.5 hour before the spawning). We did not cross-fertilization check among the colonies, but most *A. tenuis* shows high rates of crossing among colonies (Morita et al., 2018 Coral reefs). The fertilization rates often depend on genotypes of fertilization-related proteins, and thus the arguments for cross-fertilization are not crucial for this study.

Line 209: Further details needed on number of parents from which gametes were obtained for fertilization, amount of eggs/sperm mixed, and fertilization rates

Answer: Regarding our experience at fertilization in the *A. tenuis* and preparation of the gametes. We collected about 1000 eggs from each colony and mixed them (total > 4000 eggs), and then added about 10 ml sperm suspension ($1 \sim 8 \times 10^7$ sperm/ml) from each colony to the eggs suspension (1L). Finally, sperm concentration was higher than 10^6 sperm/ml. (Lines 357–358)

Line 210–211: Experiments→ Experiments

Answer: we changed it.

Line 218: Section should appear after lines 186–201

Answer: We replaced this section (Line 302).

Line 225: The four classifications of larvae (CC, CS, SC, SS) were not explained and introduced in the methodology part. You can mention those here.

Answer: Thank you for your advice. We add an explanation about classification of larvae in the first section of methods (Line 296–272) and again introduced at larvae part (Line 358–363).

Line 226: How was gamete count done? How many bundles were counted? How many eggs were measured for size? How many eggs were used for lipid analysis?

Answer: Eggs per bundles were counted for 6 bundles from 6 control and 4 stressed colonies. Egg volumes were 20 eggs from each condition (4 eggs from 4 colonies in each condition). For lipid analyses, we did only in the larvae. The number was 50 larvae each. The numbers were added in the methods section.

Line 230: Even though the method on how to measure the lipid content of the larvae is based on a paper, please provide a brief description of this method here.

Answer: We add a section about lipid extraction (Line 398–408).

Lines 233–236: Please provide relevant details on the larval experiments (how old were the larvae, what larval density, how was temperature manipulated, etc).

Answer: We started stress experiments with larvae from 5 days post fertilization (dpf) and examined the reaction for up to 21 days (26 dpf).

Line 236–237: Was this for larval size measurement? Please indicate

Answer: We add a section for larval size measurement. (Line 386–390)

Line 240–243: Please provide details on the settlement experiment. What substrate was used, what was larval density, how were settled larvae counted, how was water quality maintained over 21 days?

Answer: We add a section about settlement and size in the method. We changed the water once a day. One hundred larvae were in one container and so on. (Line 409–426)

Line 243: spats > settled juvenile corals

Answer: we edited this paragraph and edited settled planulae (Line 417 –426).

Line 246: cells > recruits?

Answer: We changed cells to recruits and extensively edited the methods section.

Figures

Line 419: “not significantly”

Answer: Thank you. We add “not” (Line 680)

Figure 3D: Instead of showing “first” and “last” on x-axis, specify the days (Day #).

Figure panels should be cited consecutively in the text.

Answer: Thank you for your advice. We changed the x-axis, and the edited figure appears consecutively.

Reviewer #2 (Remarks to the Author):

In this study by Hazraty-Kari et al., they report how a multi-year thermal stress experiment can impact reproductive output and quality of offspring from a broadcast spawning coral species. From this work, they found that heat stress increases egg number but lowers lipid content, suggesting trade-offs in gametes from the stress event. However, there was no evidence of stress hardening in the larval survival from parent colonies exposed to thermal stress. *A. tenuis* larvae and recruits already has a fairly high survival rate to heat stress compared to other species, and therefore may be more resilient to thermal events. Hence, why they are good at re-seeding the reef post mortality events.

To me, the paper needs to bolster their motivation for the study more. The experimental design is setup to address heritability or maternal effects as a form adaptation or acclimation in offspring to thermal stress of the parents. However, the authors do not touch on the large body of literature on this topic of “stress-hardening” offspring. This work is interesting because there was no evidence of improved performance of the offspring for the stressed fragments, but other differences were observed. Focusing the introduction and discussion in light of the prior papers using comparable experimental designs would enhance the future citation of this work.

Answer: Thank you for your comments. We add many section of the experimental setup and made supplementary figure 1 to show more about the adult colony, larvae, and juvenile experiments. The introduction and discussion were also edited to show more about heritability or maternal effects as a form of adaptation or acclimation in offspring to the thermal stress of the parents.

Are all offspring products of self-fertilization? The breeding scheme was not clear to me. I interpreted that all the original fragments were from one colony. Is that correct? Perhaps I missed the description of this in the methods or supplement.

Answer: The larvae were sexually produced from stressed colonies and control colonies. The mixture rates of the gametes were carefully conducted (Line 357–358).

I believe the author's need to revise the manuscript prior to publication.

Other comments:

Line 22: I would reconsider the possessive use of stress' s in this sentence. An alternative might be, "We examined the direct and carryover effects of thermal stress on fecundity..."

Answer: Thank you for your advice. We add "and" here (Line 21)

Line 23: *Acropora tenuis* should be italicized.

Answer: We changed it to litalic font (Line 23)

Line 28: I recommend removing "potentially".

Answer: We changed this sentence. "As a result of the thermal stress, we observed a trade-off in egg production; an increase in egg number but a decrease in size. In addition, larvae from the stressed colony showed marginally higher survival rates." (Line 26–28)

Line 29: As far as I can tell, symbionts were not introduced to the larvae/recruits in this study. While I agree with this statement, I do not think it should be featured so prominently in the abstract without additional context. Rather highlight the lipid differences between the treatments first and then mention how symbionts may contribute or not to larval survival. In fact, most of the results are absent from the abstract after reading the rest of the paper.

Answer: We removed this sentence and changed the sentence as shown above.

Line 57: I believe the authors mean, the larvae were released by adult colonies and then settled during high-temperature conditions?

Answer: We extensively edited the introduction including this paragraph and removed this sentence.

Line 64: This floating sentence should perhaps be combined with the previous paragraph or developed further.

Answer: We combined this sentence with the previous paragraph and show hypothesis of this study.

Line 86–87: I believe additional clarification could help the reader here that the number of eggs increased for the SF replicates that did spawn.

Answer: We did six colonies and divided them into two fragments. Four stressed fragments spawned, and six non-stressed fragments spawned (Line 112–113). In addition, we added more about experimental design in the method section and made supplementary figure 1 for readers.

Line 106: This is the first mention of planula. I would recommend either sticking with larva(e) throughout the paper or define this word that is may be uncommon in other fields.

Answer: planulae should be larvae. We changed it throughout the manuscript.

Line 235: How many larvae were used for the survival analysis? Refer to supplemental data here.

Answer: We extensively edited the method section to show more detail about the experimental design, number of colonies, and larvae for the analyses. Fifty larvae were used for the survival rate analyses in each container for control and stressed condition. We replicate these analyses for five replicates, five containers for each condition. (Line 397–406)

Line 232: How was lipid content measured?

Answer: We add lipid section. (Line 396–406)

Line 246: Cells were counted? What cells?

Answer: We apologize for this mistake. Here we man larvae. This sentence was removed and add new sections for this, “The growth rate of recruits and survival of recruits under thermal stress.” (Line 415–420).

Reviewer #3 (Remarks to the Author):

Dear authors, dear editor,

This study entitled “Adaptive strategies to increase fitness after thermal stress in the reef-building coral *Acropora*” by Hazraty–Kary et al. investigates the cross-generational acclimation potential of different life stages of the tropical coral *Acropora tenuis* to thermal stress in several short-term aquarium experiments. Adult coral fragments were exposed to a heat stress treatment and the effect on the next generation (eggs, larvae

and juvenile corals) was studied in different temperature treatments in the following year. This study claims that the heat stress treatment of the adult coral fragments improves the heat stress tolerance of the next generation.

This manuscript addresses a novel topic by investigating the effect of elevated temperature on different early life stages of corals and how temperature increases affect the next generation with a novel experimental design. The manuscript is well written and the data in the graphs are presented very well and easy to understand. The visualisation of the experimental design in Fig. 1 is very helpful for the reader. However, this manuscript is missing important information to clearly understand how the experiments were conducted and to be able to reproduce them. Partly, this information is provided in the supplementary material (but should be moved to the main text) and partly, more methodological information has to be provided (see specific comments below). In addition, this manuscript is missing detailed information from the literature in both the introduction and especially in the discussion, where the findings are not appropriately discussed. Therefore, some more background information should be provided in the introduction (incl. detailed information about the objectives, the experimental design and the research question(s) of this study) and the findings should be discussed in much more detail and in a broader context of the available literature (see detailed comments below). The discussion should rather focus on possible explanations for the main findings of this study in the context of the literature and what this means for the reproductive process of corals in the future under increasing water temperatures instead of mainly repeating the main results. With these fundamental changes in the introduction, the methods section and the discussion, I think this study will be very interesting for the audience of this journal.

Comments:

1. Please ensure consistent use of terminology in the whole manuscript and the figures as this will make it easier for the readers to understand what you are talking about. For example, you sometimes write “recruit” and sometimes “juvenile”, “control” and “normal” treatment, “oocytes” and “eggs”, “bleaching” and “brightness”, “larvae settlement” and “larvae settling”.

Title

1. I would suggest to already state in the title that this study was conducted with *Acropora tenuis* and that it focusses on different life stages of this coral species. As this is the main focus of the manuscript and a novel research topic, I would definitely highlight this already in the title, e.g. “Adaptive strategies of different life stages of the coral *Acropora tenuis* to elevated temperature”.

Answer: Thank you for your advice. We changed as you indicated.

2. I suggest to delete the key words “coral” and “*Acropora tenuis*” as they are already

mentioned in the title (or should be in my opinion as suggested above).

Answer: We deleted “coral” and “Acropora tenuis” from key words.

Abstract

3. Line 22: What do you mean by “strategies to survive for the next generation”? Please specify.

Answer: Here we mean coral may reproducing more heat-tolerant larvae. Then changed this sentence as follows; “coral may have strategies to survive by reproducing more heat-tolerable larvae.” (Line 20–21)

4. Line 23: *Acropora tenuis* in italics. Please also specify which parameters you have measured in order to be able to make statements about the fitness of the corals.

Answer: Thank you for your advice. We changed italic Acropora tenuis. We mentioned gamete parameters (egg number and size), the larval stage (survival rates and size), and recruits (survival and growth rates). (Line 25–26).

5. Line 24: Please include the duration of the experiment.

Answer: we exposed fragments to high water temperature for 10 days.(Line 26)

6. Line 26: State which parameters were measured for the juvenile corals. As you only investigated them for 21 days, I would suggest to rather call them coral recruits instead of juveniles (as you also sometimes do in the manuscript).

Answer: We added parameters of larvae and juveniles. And we also changed juveniles to recruits (survival rates and growth). (Line 27 – 28)

7. Line 27: It is not clear why you explicitly state that you only investigated the “stressed corals” as the control corals were also investigated I guess, please specify.

Answer: We changed these sentences as follows to describe more precisely. “As a result of the thermal stress, we observed a trade-off in egg production; an increase in egg number but a decrease in size.” (Line 26-27)

8. Line 28: You do not state any results that support your statement of “increased larval production” as you only examined the “egg production”. Is this what you mean here? Please clarify.

Answer: Thank you for your suggestion. Colonies produced more but smaller eggs after they were stressed, and the larvae from stressed colonies were marginally heat-tolerant. We changed these sentences as follows; “In addition, larvae from the stressed colony showed

marginally higher survival rates. Collectively, heat-vulnerable corals have strategies to produce more heat-tolerant larvae after they experience stress.” (Line 27-30).

9. Lines 28ff: The statements of the main results are not very clear in the abstract: e.g. “potentially higher rates of survival for larvae and juveniles”, but you found highest survival rates of recruits in control treatments (lines 129ff). Or do you mean the higher survival rate in SS compared to CS here (line 133)? Which results support your statement of “Symbiotic relationships may assist the survival of larval stages with lower lipid content”? Please state the results more clearly in the abstract.

Answer: Thank you for your suggestion. We deleted about symbiotic relationship statement. About recruits survival rates, as you indicated highest survival rates were CC condition, but we would like to emphasize that larvae from the stressed colonies (SS) showed more heat tolerance than those from the control condition (CS). Then, we changed these sentence as follows; “In addition, larvae from the stressed colony showed marginally higher survival rates. Collectively, heat-vulnerable corals have strategies to produce more heat-tolerant larvae after they experience stress.” (Line 27-30)

Introduction

10. As you investigate the effect of elevated temperature on adult corals and early life stages, please provide some more general background information in the introduction about what is known so far about the effect of increasing temperature on different life stages (larvae, recruits and adults) of corals to make more clear what is novel about this study. I include some references that I think might be useful to include in the introduction and which can also be interesting for the discussion. Even though the study by Putnam & Gates (2015) investigates the trans-generational acclimatisation of corals under elevated temperature and pCO₂, I think it is still important to include this study in this manuscript to show what has already been done in previous similar studies.

Anlauf, H., D’Croze, L., & O’Dea, A. (2011). A corrosive concoction: The combined effects of ocean warming and acidification on the early growth of a stony coral are multiplicative. *Journal of Experimental Marine Biology and Ecology*, 397, 13–20.

Bahr, K. D., Tran, T., Jury, C. P., & Toonen, R. J. (2020). Abundance, size, and survival of recruits of the reef coral *Pocillopora acuta* under ocean warming and acidification. *PLoS ONE*, 15, e0228168.

Chua, C. M., Leggat, W., Moya, A., & Baird, A. H. (2013). Temperature affects the early life history stages of corals more than near future ocean acidification. *Marine Ecology Progress Series*, 475, 85–92.

Foster, T., Gilmour, J. P., Chua, C. M., Falter, J. L., & McCulloch, M. T. (2015). Effect of ocean warming and acidification on the early life stages of subtropical *Acropora spicifera*. *Coral Reefs*, 34, 1217–1226.

Jiang, L., Zhang, F., Guo, M. L. et al. (2018). Increased temperature mitigates the effects of ocean acidification on the calcification of juvenile *Pocillopora damicornis*, but at a cost.

Coral Reefs, 37, 71–79.

Liberman, R., Fine, M., & Benayahu, Y. (2021). Simulated climate change scenarios impact the reproduction and early life stages of a soft coral. *Marine Environmental Research*, 163, 105215.

Negri, A. P., Marshall, P. A., & Heyward, A. J. (2007). Differing effects of thermal stress on coral fertilization and early embryogenesis in four Indo Pacific species. *Coral Reefs*, 26, 759–763.

Putnam, H. M., & Gates, R. D. (2015). Preconditioning in the reef-building coral *Pocillopora damicornis* and the potential for trans-generational acclimatization in coral larvae under future climate change conditions. *The Journal of Experimental Biology*, 218, 2365–2372.

Randall, C. J., & Szmant, A. M. (2009a). Elevated Temperature Affects Development, Survivorship, and Settlement of the Elkhorn Coral, *Acropora palmata* (Lamarck 1816). *Biological Bulletin*, 217, 269–282.

Randall, C. J., & Szmant, A. M. (2009b). Elevated temperature reduces survivorship and settlement of the larvae of the Caribbean scleractinian coral, *Favia fragum* (Esper). *Coral Reefs*, 28, 537–545.

Ross, C., Ritson-Williams, R., Olsen, K. et al. (2013). Short-term and latent post-settlement effects associated with elevated temperature and oxidative stress on larvae from the coral *Porites astreoides*. *Coral Reefs* 32, 71–79.

Schnitzler, C.E., Hollingsworth, L.L., Krupp, D.A. et al. (2012). Elevated temperature impairs onset of symbiosis and reduces survivorship in larvae of the Hawaiian coral, *Fungia scutaria*. *Mar Biol* 159, 633–642.

Answer: Thank you for your suggestion. We referred the suggested references and extensively edited and added new introduction parts. Please see the paragraphs (Line 41–91 in red characters).

11. As this manuscript investigates different coral life stages, please provide some information about the life cycle of *A. tenuis*.

Answer: We add this information in the first sentence of the methods section. “we used the reef-building coral *Acropora tenuis*. The coral *Acropora* is a broadcast spawner. Larvae disperse after fertilization within released gametes and recruit in the reef.” (Line 264–265)

12. Line 42f: I think it would be helpful for the reader to provide more information from the literature about the different mechanisms how corals can acclimate to elevated temperatures.

Answer: We predict coral can acclimate to higher water temperature via tolerance of bleaching without reduction of photosynthetic activity, growth rates, and survivorship. We newly introduced about your suggestion in the introduction (please see red characters in line 41–90).

13. Line 43: Can you include examples of more heat-tolerant species?

Answer: There are many species can survive the higher water temperature, we cite the good example of our study sites before and after heavy bleaching. Then, indicated one massive species as an example (*Porites lutea*). (Line 44)

14. Line 45: “Physiological performance” is a very general term. What do you mean exactly? Could you please specify and also provide some more recent references here?

Answer: Here we mean photosynthetic activity and respiration, and added references. (Line line 45–48)

15. Line 48: I would delete “young corals” as you also write “early life stages”.

Answer: thank you for your suggestion. We deleted “young corals” and extensively edited this paragraph..

16. Line 55f: You state that “the thermal stress history of parents can influence the thermotolerance of early life stages in corals and other marine organisms (30, 31)”, but ref. 31 is an acidification experiment. Please provide another suitable reference here.

Answer: Thank you for your advice. We changed ref 31 Moya et al., to 17 Putnam et al., 2015.

17. Line 57f: I do not understand this sentence “Thus it is highly likely that···, resulting in high-temperature conditions” as not the coral larvae lead to elevated temperature, please rephrase.

Answer: We deleted this sentence, and we rewrote most introduction parts about early live stages of the corals (Line 55 – 90).

18. Line 58f: Can you provide more information about how much the temperature fluctuates in the habitat of *A. tenuis*? As many tropical coral species are experiencing high temperature fluctuations, is this species subject to especially high temperature variability?

Answer: We could not precisely present the water temperature in Okinawa, but added the several references about the thermal stress on the corals (Line 56–67). In addition, we also discussed about the water temperature and the coral reef in the discussion “Therefore, the larvae need to grow near the habitat of the parent colonies, and the water temperature in the summer of their habitat potentially becomes higher. In the case of the *Acropora tenuis*, their habitat is shallow (<10 m) and primarily influenced by heat waves in the summer⁶⁵.” (Line 239–242)

19. Line 60f: Please specify how responses to elevated temperature vary across species.

Answer: We added several arguments in introduction about the thermal stress in several species “The adult reproductive output as egg production varies after thermal stress. The effect of heat stress is often negatively affected in many coral taxa, and the number of eggs becomes smaller^{21,28}. In addition, the fecundity of the colony also declines due to the smaller number of polyps producing eggs²⁸. Although thermal stress is implicated in changes in egg sizes, the trade-off between eggs size and fecundity is not clearly shown²⁹.”(Line 49–54). In addition ,in the following paragraph, including *Acropora* and the other genus. (Line 55–67)

20. Line 61f: References missing for “...the effects of thermal stress on adults, larvae, and juveniles have been reported in previous studies”.

Answer: We added references in this sentence (Line 80).

21. Line 64f: Please provide more information about the objectives of this study in the last paragraph of the introduction as this is not clear yet, incl. a short statement of the research questions/objectives you tried to answer with this experiment, a short summary how the experiments were conducted (experimental design, referring to Fig. 1) and which parameters were measured. I would also suggest to state here that this study was conducted with *A. tenuis*.

Answer: We add information citing references and show hypothesis and objectives in this paragraph as follows; “In a higher water temperature, larval development become faster^{20,34} and metamorphosis also occurs faster³⁵ but also show higher mortality^{20,34,36}. As a result of faster development of the larval stages, they tend to settle near the adult colony³⁷. The larval dispersal becomes smaller range, and the number of the recruited larvae could increase after the degradation of the reef due to heavy bleaching event. From this aspect, more heat tolerance larvae are desirable in a higher water temperature and will be a good strategy for their fitness after colony experiences heat stress.” (Line 60-66)

Methods

22. In my opinion, the methods section is missing some important information to fully understand how the experiments were conducted. I would suggest to move all the information that is already provided in the supplements, but is essential to fully understand the experimental design, to the main text.

Answer: We extensively edited the methods section to show more in detail about experiment.

23. Line 185: I suggest to shorten subheading to “coral collection and experimental design”.

Answer: Thank you for your advice. We shorten the subheading according to your comment. (Line 263)

24. Line 186: When was the experiment conducted?

Answer: We started in 2018. Eggs, larval, and recruit experiments were in 2019. We added information when we started the experiments in the method section. (line 273–301)

25. Line 187: I suggest to replace “adults” by “adult physiology” (or more precise term) as you also state that you measured “adult fecundity” to make the difference between the two more clear.

Answer: Thank you for your comment. We changed “adults physiology”. (Line 267)

26. Line 189: How many fragments were used? How many replicates in how many separate aquaria?

Answer: We used six colonies and divided them into two fragments for control and stress conditions. Total, six fragments for control and stressed treatment respectively. We added this information in supplementary figure 1 and lines 276–277. “All colonies were sectioned into two fragments and were tagged as two different treatments (control and stress treatments).”

27. Line 192: How was ensured that the temperature in the control treatment was stable during the experiment when a heater was only placed in the heat stress treatment? What was the water flow in the flow-through system? How were constant conditions ensured with the constant flow of water?

Answer: We used to flow through the water tank and measured water temperatures, and so on. All this information was added in the methods section. (Line 277–295)

28. Could you please provide photographs of the aquarium system in the supplements to better understand the setup of the aquarium system?

Answer: We made a new supplementary figure 1, showing the requested information.

29. Line 193: Please state the exact mean \pm standard deviation for both temperature treatments. Also mention how and how often the temperature was measured during the experiment. Please explain why you decided to conduct a $+3^{\circ}\text{C}$ heat stress treatment. Why did you decide to conduct the experiment at 31°C and not at a higher or lower temperature? Was this decision based on the temperature range that the corals experience in situ or that they will experience until the end of this century? Then please provide evidence for this.

Answer: We added the mean and standard deviation of the measured temperature ($27.5 \pm 0.02^{\circ}\text{C}$ on average (control) and $31.2 \pm 0.01^{\circ}\text{C}$ (stress) Line 279-281). At the experiments, we treated 28°C as a control condition corresponding to 1°C lower than the average maximum monthly temperature and applied $+2$ degrees of the average maximum monthly temperature. This stress condition is slightly less than the bleaching condition ($20\text{ DHD} < 3\text{ DHW}=21\text{ DHD}$).

30. Did you also measure other seawater parameters such as salinity, oxygen concentration and pH during the experiment and could you please also provide this information (mean +/- SD)?

Answer: Unfortunately, we did not measure other parameters such as salinity and oxygen concentration, and so on. We added mean and SD information for temperature (Line 280–281, 363–365).

31. Line 184f: What exactly do you mean by “until the fragments started to show visible bleaching”? Please provide some more information here.

Answer: We apologize for this sentence. In truth, we did not observe any bleach during heat-stressed conditions. I am Masaya Morita, one of the corresponding authors who misunderstood this part and added these words. I apologized on behalf of the authors and erased these words from the sentence.

32. Line 195f: How was the 12-h light:12-h dark cycle controlled? Were the corals only kept in the outdoor tank for the acclimation period as mentioned in the supplements or also during the experiments?

Answer: We made supplementary Figure 1. The light was controlled with the timer. Corals for acclimation periods were in the aquarium in the outside tank and corals were kept in the indoor tank set light and heater.

33. Line 197f: As you regularly took pictures of the corals, I think it would be interesting for the readers to see pictures of these corals after the heat stress experiment to see the degree of bleaching (see my comment above). Please provide some pictures of the corals in both treatments as examples.

Answer: Thank you for your advice. I mentioned above that the coral fragments were not bleached, but showed many pictures of the corals during the experiments in the supplementary Figure 1.

34. Line 199f: “Physiological parameters (including the symbiont algal density and chlorophyll content)”: Which other measurements were conducted if you write “including”? When exactly were the measurements conducted? How many days before/after the experiments? Do you also have data about the growth or respiration rates of the corals over the time of the heat stress treatment and under control conditions?

Answer: We measured color, symbiotic algae density, and chlorophyll content. Then, we deleted “including” from this and added “color”. (Line 298). We applied stress for 10 days, and unfortunately, we did not measure respiration rates. We add detailed methods in

the following physiological and survival rates measured 10 months after the stressed application (Line 341–345).

35. Line 200f: Please state why you did not bleach the corals right before the spawning event but decided to bleach them one year before the spawning. This is not clear but important to understand the reasoning for this part of the experimental design.

Answer: Apologies again. Stressed coral fragments were not bleached and just exposed to thermal stress. The water temperature during the spawning month is far below bleaching condition (25–26 °C), and thus we did not apply the stress. The reason why we measured physiological parameters was to estimate the degree of stress.

36. Line 202f: When exactly were the corals re-collected in spring 2019, when were the experiments conducted and when did the spawning event occur?

Answer: We added this information in the method section. Spawning occurred on June 12 2019 (Line 347–351).

37. Line 206f: Here you state that you used two treatment conditions after re-collection of the corals but experiment 2 in 2019 seems to have been only conducted under control conditions in Fig. 1 (blue aquaria), please clarify.

Answer: Spawning condition, we did not expose the fragments to stress and control conditions. In this study, we plan to examine the effect of heat waves in the summer. The spawning month is still cooler than in the summer, and thus we did not examine its effects on the spawning. The larval can survive for a longer duration and recruits need to pass the summer.

38. Line 208: How many aquaria and how many fragments in each aquarium were used?

Answer: We separated the fragments before the spawning. We used six fragments of the control and five fragments of the stressed one. At the spawning 11 aquariums were used to collect gamete bundles.

39. Line 218: When were the physiological parameters of the adult corals measured? I assume in 2018. Therefore, I would write this paragraph before you describe the experiments in 2019 in order to describe the experiments in chronological order (and the same order as in Fig. 1), which will make it easier for the readers to follow.

Answer: We add information about physiological parameters in adult corals (color-brightness, maximum photosynthetic quantum yield-Fv/Fm, symbiotic algal density, and chlorophyll a and c2). (Line 302–340)

40. Line 232: When were the larvae for the lipid analysis sampled and how were they processed? I assume also according to ref. 44?

Answer: we added information about lipid analyses in the method section (Line 398–408).

41. Line 234ff: Please describe the setup of the glass containers in more detail: How were the five glass containers set up? How was water movement inside the containers ensured? How was the water replaced without losing any larvae? How many larvae were in each glass container? How did you measure the water temperature inside the containers and how did you ensure stable temperatures? In Fig. 1 it looks like the glass containers were placed into aquaria, were the five containers placed in the same aquarium or in separate ones? Where did the larvae settle on? In Fig. 1 it looks like the larvae settled on small plates that were placed in the aquaria afterwards, is this correct? Please provide more information here. You write “five replicates x two treatments” but according to Fig. 1 you had four treatments (CC, CS, SC, SS), please clarify. Please also state in the text what the abbreviations CC, CS, SC, SS stand for as this is only partly explained in the figure legends.

Answer: we add information about your questions in the method section (in red letters). Water movement was not measured, but the aquarium was set up inside the building under LED illumination, and thus the water temperature could be stable. In addition, we made supplementary figure 1 to show more detail about the aquarium setup. Five replicates mean five containers of larvae from control or stressed fragments for each treatment (stressed or control) (for CC, CS, SS, and SS). (Line 391–426)

42. Line 237ff: Please state why you took pictures of the larvae and also provide example pictures of the larvae of the different treatments if you observed any differences.

Answer: We took photographs of the larvae to measure the size of the larvae. (Line 409–416).

43. Line 240ff: Please state that the settlement rate was investigated over 2 days. How was the size of the settled planulae/recruits measured, what did you measure exactly?

Answer: We added information about settlement behavior and size of the larvae in the method section, and described in detail what we did (Line 409–422).

44. Line 246: It is not clear what you mean by “the cells were counted”, do you mean the number of early recruits? Could you also provide pictures of the young recruits for the different treatments?

Answer: Yes. We counted the recruits and added photos of the early recruits in supplementary figure 1. We apologize for not to describe well.

45. Line 258: Which distribution of the data was used for the generalized linear model, i.e. which “family” in the glm function?

Answer: we used gaussian distribution for the GLM analyses.

Results

46. Line 69f: What is meant with “brightness”? How is this measured? As you apparently mean “bleaching” here, I would change the term in the text to make this more clear. As you observed significant differences between the treatments, please also provide some pictures of the coral fragments.

Answer: Here we mean here is the color scale. A detailed description was added in the methods section, Coral color measurements (Brightness) (Line 303–314).

47. Line 70f: I would suggest to delete the sentence “We also measured…” as this was already described in the methods section.

Answer: We deleted this sentence.

48. Line 76: I think it is a bit confusing to talk about “highest” here as you only have two treatments, please rephrase.

Answer: We changed “highest” to “higher”. (Line 101)

49. Line 79f: Do you mean Chl a here?

Answer: Yes. Thank you for your suggestion. We add a in here. (Line 105)

50. Line 81f: You state here that the chlorophyll c_2 content was significantly different but this is not shown in Figure 2F, please correct.

Answer: Thank you for your suggestion. We changed Fig.2F to Fig. 2d and e (Line 107)

51. Line 83f: Please add here the information over which time period the survival was measured.

Answer: We added information about the duration of survival we measured, three months. (Line 108–110)

52. Line 86f: The sentences “One SF did not spawn at all during the spawning period. However,…” are not clear as you talk about just one fragment in the first sentence but about all fragments in the second sentence, I assume. I would suggest to rephrase these sentences and delete “however”.

Answer: We changed this sentence as follows; Four colonies in stressed (SF) and six of those in the control treatment (CF) spawned. One colony of the stressed condition (SF) did not spawn. (Line 112-113)

53. Line 95: “SF larvae exhibited substantial thermal tolerance”: How does this statement fit with the results as the larvae in both thermal stress treatments had lowest survival rates.

Answer: We deleted the sentence.

54. Line 96f: The survival rates of the treatments CS and SS are exactly the same, is this correct?

Answer: We apologize for this, this is mistake We changed these values as follows; CS 82.6 +/- 1.6 % SS 85.0 +/- 1.5 % (Line 134)

55. Line 102: I guess you mean “Supplementary Table 1” here?

Answer: thank you for your suggestion. Yes. We added supplementary here. (Line 139)

56. Line 103: The sentence “To estimate larval size,···” should be stated in the methods section and can be deleted here.

Answer: We deleted this and added more about larval size in the method section. In addition, this section was moved to the first part of the larval response (Line 121–124).

57. Please provide a table with all the results of the statistic tests (not just the p-values) in the supplements, not only for the results of the survival analyses.

Answer: We made supplementary tables 1 to 6 for statistics.

58. Line 121f: The sentence “In each treatment,···” should be stated in the methods section and can be deleted here.

Answer: We deleted this sentence and added more in detail about this in the method section.

59. Line 123ff: Here, it is stated that the CC recruits are significantly larger than CS, SC and SS, but this is not shown in the figure. Please also include the statistical results in the figures and do not only show the significant results of CC–SC.

Answer: Thank you for your advice. We changed figure 4b for all comparisons.

Figures

60. I would suggest to use the same four colours in Fig. 1 for experiments 3 and 4 as in

figures 3 and 4 (incl. light blue for CS and pink for SS) and dark blue for CF for experiment 1 and 2 to make it more clear for the readers. I would also use the four colours in Fig. 4B instead of just red and blue.

Answer: Thank you for your suggestion. We changed all colors.

61. Line 401f: The control group was also maintained in the aquarium for 10 days, correct? Please clarify.

Answer: Yes, we add information about this as follows; The Control group was also reared in the control condition for 10 d.(Line 658 - 659)

62. Line 403: I would suggest to write “larvae settlement” instead of “larvae settling” (also in the figure) as in other papers and other parts of the manuscript.

Answer: We changed “larvae settling” to “larval settlement”. (Line 660)

63. Line 404f: But the juvenile corals of the control treatments were also investigated, but this is not stated here, please clarify.

Answer: Thank you for your advice. We changed here as follows; Effects of thermal stress on recruits obtained from stressed and control fragments.(Line 661)

64. Line 407f: What is meant with “brightness”? How is this measured? As you apparently mean “bleaching” here, I would change the Y-axis label and the term in the figure legend and in the text. I would suggest to write “survival” instead of “mortality” as this is also written in the figure. As you also plot the symbiont density, Chl a and Chl c₂ here but this is not mentioned in the figure legends, maybe change the figure legend to “Effect of thermal stress on the physiological performance and survival of adult coral fragments (Experiment 1)” or something similar. I would also mention somewhere in the figure legend that these data are from *Acropora tenuis* as figure legends should be understandable without reading the main text.

Answer: Thank you for your advice. We revised figure legends to show more in detail about the content.

65. Line 409f: How did you determine the Y-axis values of 50 and 225 for healthy and bleaching? And what is the unit for the Y-axis?

Answer: We added a description of these values in the methods section of coral color measurements (Brightness) (Line 303–314).

66. Line 413ff: I do not understand the survival graph of Fig. 2C. According to the figure legend you had 6 fragments per treatment, but according to the graph 8 fragments died

over the 10 months but this only represented about 25 % of the fragments.

Answer: The figure was changed to show the results precisely; please see new figure 2f. Six of five colonies survived for 10 months.

67. Figure 3: Please specify in the figure legend how many fragments per treatment were used to count the number of eggs in the bundles in Fig. 3A. It sounds as if $n = 40$ fragments were used in Fig. 3B but if I understood it correctly, these are the number of egg bundles, correct? Please clarify. Why were 30 bundles used for Fig. 3A but 40 bundles for Fig. 3B? What does “first” and “end” on the X-axis of Fig. 3D mean? Please specify in the figure and the figure legend as this is not clear when the lipid concentration was measured. On the X-axis of Fig. 3E are two times 0 days instead of 0 and 5 days. In Figure 3E, I suppose you had four treatment groups in the end. I think it is a bit confusing to talk about “two treatments” here.

Answer: We extensively revised figures and legends. We measured 6 bundles from each colony and total 24 bundles from 4 stressed colonies and 30 bundles from control colonies. 40 eggs were a mistake, and we changed this part. In the lipid amount of “first” and “end”, this means days started and finished. We changed the first to 0 and the end to 21 (Fig. 3e). About the survival rates of the larvae, we apologize for the mistake. The date was changed, and four treatments were shown with legends of the treatments (CC,CS,SC, and SS). Thank you for your valuable comments.

68. It is not clear what the settlement size in Fig. 4B is. Is this the diameter or height of the coral recruit or something else? How was this measured? Please specify in the methods section and in the figure legend.

Answer: We measured the largest diameter of the recruits, and added the new methods section named “Settlement behavior and size” in the method section.

69. Do you have any information about the growth rates of the coral recruits during the 21 days after settlement? As you took pictures of the coral recruits, I would suggest to also measure their growth and include this information in the manuscript as this is a very important trait in my opinion. Even if most recruits were able to survive the first 21 days after settlement in the four treatments, there might be differences in growth rates that are important to also take into account for the development of the coral recruits.

Answer: We added information about the growth rates of the recruits for 60 days (in Fig. 4C). In addition, the results of the growth rates of the SS recruits were significantly larger than that of CS.

Discussion

70. In my opinion, a thorough revision of the discussion and conclusion is necessary and I will therefore only comment on general things and not on details at this point. Please

focus less on the results of your experiments (which are already stated in the results section) and more on (potential) explanations for your findings, do not just repeat the results here as you do in large parts of the discussion. Please discuss your findings in much more detail instead. In cases where no other studies are available yet, I would like to see some speculation about your findings and what you think this means for the corals. Please also discuss all findings of this manuscript (all measured traits) and if you think that the findings of some of the traits are not important enough to be discussed, then please do not show the results in the manuscript or move them to the supplements (e.g. the physiological data of the adult coral fragments are not discussed at all, whereas a figure showing the lipid content depending on egg size is missing).

Answer: We extensively edited the discussion to answer the comments.

71. Please discuss that most results of the larvae and recruits do not seem to be related to the heat stress treatment of the adult corals (Experiment 1) but rather to the heat stress treatments in Experiments 3 and 4. I think this is a very important point and should be discussed (also see comment below to Fig. 4E).

Answer: We added a discussion about the thermal stress responses of the larvae and recruits. We added in the discussion about non-significance in larvae and recruits between control and stressed condition. "In addition, the larvae and recruits between control and stressed fragments were not significantly different in the control condition. Therefore, parents may prepare eggs against heatwaves after the subsequent spawning." (Line 254-256)

72. As you mention in the introduction, marine heat waves will increase in the future and corals will experience warmer temperatures all over the world. I think it is therefore very important to also discuss what the results of your study mean for the reproductive success of adult corals and the development of larvae and recruits in the future in a warmer ocean in more detail.

Answer: Thank you for your comments. We added a new paragraph in lines 235-247.

"Larvae and recruits from the stressed fragments show slight heat tolerance than those from non-stressed control, implying that their survival at these stages is reasonable for their survival and reef maintenance. In this study, larvae showed more rapid settlement rates in the higher water temperature (Fig. 4B CS and SS). Like in our study, the higher water temperature causes restriction of larval dispersal due to rapid larval development and settlement^{22,31,41,42}. Therefore, the larvae need to grow near the habitat of the parent colonies, and the water temperature in the summer of their habitat potentially becomes higher. In the case of the *Acropora tenuis*, their habitat is shallow (<10 m) and primarily influenced by heat waves in the summer⁶⁶. Although the frequencies of the heat waves are not considered in this study (exposed once ten days a year), the stressed colony could produce more slightly heat-tolerant larvae. This acclimation potential is helpful for their population. Still, reproductive success after spawning is limited when the number of colonies declines due to heavy bleaching. In addition, the juvenile could not survive when the summer heatwaves frequency was high (Hazrati et al., in preparation)."

73. Line 149: Where are the results shown that "the lipid content of coral eggs differed with egg size?" I did not find any figure that shows this or any information about this in

the results section.

Answer: We apologize for this mistake, we did not measure the lipid content of the eggs but measured that of the larvae. The lipid content of the larvae from the stressed fragment (SC and SS) was smaller than those of the control fragment (CC and CS) (Fig, 3e).

74. Line 158: But the survival rate of larvae seems to depend more on the heat stress treatment in experiment 3 (as shown in Fig. 3E) than the heat stress treatment of the adult coral fragments. How do you come to the conclusion that the larvae of the stressed coral fragments were not vulnerable to thermal stress?

Answer: Here we means the larvae from the thermally stressed fragments were more vulnerable in the stressed condition, not in the control condition.

75. Line 160: Same as previous comment: How do you come to the conclusion that the SF juveniles were more tolerant to high water temperatures? I think if you discuss this here that SS were slightly more tolerant than CS, then you should also discuss that they were nevertheless both less tolerant than CC and SC. The response of larvae and juveniles seems to depend more on the heat stress treatment in experiments 2–4 than in experiment 1.

Answer: Thank you for your comment. We discussed these points in the newly added paragraph (line 235–247) and sentences “In addition, the larvae and recruits of the survival rates between control and stressed fragments were not significantly different in the control condition. Therefore, parents may prepare eggs against heatwaves after the subsequent spawning.” (Line 253–255).

76. Line 176ff: In my opinion the strong statement “Corals also exhibited thermal acclimation under thermal stress. The acclimation potential of a subsequent generation identified in this study indicates the possibility that future generations may survive conditions of thermal stress” is not supported by the results of this study as survival rates only slightly increased in SS compared to CS, please rephrase and generalise less.

Answer: We deleted this sentence. The former sentences are more appropriate and not necessary this sentence. “We also found that colonies that experienced heat stress developed tactics to increase larvae and recruit survival in higher water temperatures. For example, colonies under heat stress produce smaller eggs in larger numbers, which could increase larval settlement and lead to reef maintenance. In addition, the larvae and recruits of the survival rates between control and stressed fragments were not significantly different in the control condition”(Line 251–255)

Supplementary information

77. “Four different life stages of corals: adults, fecundity and eggs, larval settlement, and early recruits.” However, “fecundity” and “larval settlement” are not life stages of the corals. I would suggest to either only state the life stages that you investigated or the

parameters that were measured.

Answer: Thank you for your advice. We moved to the method section and described it as follows; “In this study, we used the reef-building coral *Acropora tenuis*. The coral *Acropora* is a broadcast spawner. Larvae disperse after fertilization within released gametes and recruit in the reef. Here, we examined the effects of thermal stress on the following life stages and functions in the coral *Acropora tenuis*: adults **physiology** (Experiment 1), adult fecundity (egg number and size) (Experiment 2), larval **survival and** settlement rates (Experiment 3), and recruits (Experiment 4) (Fig. 1).” (Line 264–269).

78. “...until the fragments started to show a visual bleaching response” and “The measurements were repeated every day until the signs of bleaching were revealed approximately ten days after the thermal stress”: Was the experiment conducted for exactly 10 days as stated in the main text or did the experimental duration differ for each fragment? Or did it just differ when the fragments started to bleach? In that case, when did you decide to stop the experiment? Were all fragments bleached after 10 days? This is not clear, please clarify.

Answer: We moved this section to Methods. We applied heat stress for 10 days and brightness was changed, but bleaching did not occur. (Line 283–284).

79. It is not really clear how the surface area of the corals was measured, please rephrase this sentence. I assume that the surface area was determined based on the paraffin wax dipping technique. How did you determine the “mean surface area” for each fragment? Did you repeat this procedure several times as you write that only one branch of each fragment was used for this analysis?

Answer: We measured with surface area with Veal et al 2010, We add information about this in the method section. “Finally, following the standard procedure, the algae's mean total number per mean surface area of each fragment was divided into treatments based on the paraffin wax dipping technique⁶⁷” (Line 332-334)

80. Why was only one branch per fragment used to determine the algal symbiont densities and chlorophyll a and c_2 content? Do you have any information about the variability of these parameters in the same coral fragment or in *A. tenuis* in general?

Answer: We cut only one branch and added a reason in the method section “Only one branch was collected to avoid escalating stress and keep the fragments surviving until the subsequent spawning in 2019.” (Line 326-327). Honestly, we cannot estimate the variability of the obtained data using only one branch. However, the one branch contains many polyps and estimated the measuring using only one branch is enough.

Reviewers' comments:

Reviewer #1 (Remarks to the Author):

The authors have addressed most of the reviewer comments and suggestions by adding relevant context for their study in the introduction, and by discussing their results in comparison to findings described in previous studies. The methods section now has more detail, although some information is still lacking.

The paper is markedly improved content-wise and conclusions are appropriately stated. However, I recommend that the manuscript undergo extensive editing to improve clarity of the statements throughout and to reduce redundancy, especially in the introduction and discussion.

Specific comments:

Title

Line 1: "...life stages of the coral..."

Intro

Line 41: "photosynthesis" instead of "photosynthetic activity"

Line 45: may bring what? sentence is incomplete

Line 56-57: not clear what the relevance of life stage length is in this paragraph

Line 61-62: please rephrase for clarity

Line 63: "decreases" instead of "declines"

Line 76: is "consecutive" correct?

Results

Line 94: First paragraph of results needs a short introduction to what was done to test physio responses of adult colony fragments

Line 108: "one fragment from the stressed treatment died 3 months after the thermal stress"

Discussion

Line 189-190: does this mean that bleaching was observed in the treatments for the adult colony fragments?

Line 194-196: not sure what is meant by this statement

Line 220: statement on growth of marine invertebrates seems out of place in a paragraph discussing lipid utilization

Line 223: are there other studies where slower rates of lipid utilization were also observed at elevated temperature?

Line 224: could the decrease in lipid utilization be because the larvae are not healthy at higher temperature?

Line 225-226: what is meant by this statement?

Line 230-243: not sure what is meant by these statements; please clarify

Line 235: "showed slighted higher heat tolerance"

Line 236: rephrase for clarity

Conclusion

Line 253: "...survival rates of larvae and recruits from control and stressed fragments..."

Line 255: what is meant by "parents may prepare eggs against heatwaves after the subsequent spawning"? please rephrase for clarity

Methods

Line 402: were larvae at 21 days still swimming? Were these setups separate from the ones for settlement? was a settlement cue or substrate added into these setups?

Line 410: how old were the larvae when they were placed in the settlement setups?

Line 417: were larvae maintained at high temperature for up to 2 months prior to measurement of size?

Reviewer #2 (Remarks to the Author):

The authors have addressed my prior concerns through the addition of text within the introduction on the motivations behind the study and background literature in both the introduction and discussion to provide context to their results. The text still requires polishing, especially the new text, before it can be published.

Below are my new comments on the revised manuscript.

Second Paragraph of Introduction: I found the two comparisons in this paragraph to be a bit confusing as currently written. What I think you are saying is, some corals like *Porites lutea* can withstand thermal stress unharmed but other species, like {fill in blank} cannot. They experience declines in physiological performance such as those referenced starting on Line 45.

Line 44-45: The sentence starting with "However, the higher..." is incomplete.

Line 49: Do you mean adult reproductive output "as measured by" egg production varies?

Line 49-50: I struggled through the first part of this sentence. Perhaps you might change it to, "The effect of heat stress on reproduction is often negative in many coral taxa" or a shorter version "For many coral taxa, the effect of heat stress reduces the number of eggs." When you refer to the eggs become smaller, are you talking about total number of eggs released or the size of eggs released decreases?

Line 53: Change "between egg size and fecundity is" to "between egg size and fecundity has not been".

Line 53: I would consider changing larval traits derived from stressed colonies to the inheritance of larval traits from stressed coral colonies.

Line 62: Do you mean "larval survival" where you have written larval stages?

Line 68: I was a bit lost at the comparisons at the end of this sentence. Please revise.

Line 75: "This study intended.." (past tense).

Line 76: I would consider changing how this sentence is written. For example, "to reveal how consecutive heat stress on the adult corals effects the larval and juvenile performance under similar stress conditions."

Line 85: more heat "tolerant" larvae

Line 94: Introduce brightness as a metric for bleaching. For example, "First we measured the brightness of the coral fragments, a color scale that corresponds to symbiont loss "

Line 114: The mean+- standard error of the mean, SE can be removed here as it is also presented on Line 101.

Line 140: I think you mean settlement size rather than larval size.

Line 193: "Lepstastrea is not phylogenetically related to the genus Acropora,"

Line 199: Is there a reference for this statement?

Line 217: I am a bit confused here. You are unsure if any larvae in this experiment are symbiotic, so are you saying that this might be a strategy for survival in the field (outside the scope of this experiment). Please clarify.

Line 235: Larvae and recruits from the stressed fragments "showed" slight heat tolerance "compared to" those from non-stressed control.

Line 235: Only the SS recruits were sig. different from CS recruits, right? The larvae did not show differences in heat tolerance (no sig. differences).

Line 246: juveniles

Reviewer #3 (Remarks to the Author):

Dear authors,

thank you for thoroughly revising your manuscript. The results are stated much clearer now, the introduction gives a better overview of the state-of-the-art, the methods are stated more clearly and it is much easier now to fully understand the experimental design and you discussed your findings in more detail. However, it is a pity in my opinion that no other environmental parameters than water temperature were measured during the experiment to ensure that only the temperature differed between the treatment conditions (both in experiment 1 and experiments 2-4). As all aquarium systems in experiment 1 were supplied with water flow-through, I assume that all other parameters were the same in both treatments in this experiment, but we cannot be sure for experiments 2-4 and I think it would be important to clearly state this limitation in the manuscript. I still noticed some inconsistencies in the manuscript (see my detailed comments below) that should be remedied, in my opinion. I am particularly concerned that the temperature of the two control treatments (CC and SC) differs by more than 2 °C as such a large temperature difference will most likely affect the development of the larvae and it cannot be assumed that they were exposed to the same "control" conditions. Therefore, I would like the authors to give a reason why this temperature difference occurred, clearly state this information in the manuscript (not only in the methods section) and all figure captions and include this in the discussion of their results.

Abstract:

1. I do not agree with the strong statement that "heat-vulnerable corals have strategies to produce more heat-tolerant larvae after they experience stress" (lines 28-30) as your results show that the larvae of the control treatment were larger, had higher lipid content and had higher survival rates than the heat-stressed larvae (lines 121-139). Please clarify here.

Introduction:

2. Line 41-48: This paragraph is a bit confusing because you first state that corals might be able to acclimate to elevated temperature, but then you write that elevated temperatures will decline their physiological performance. Please state the state-of-the-art more clearly and why different studies found different responses. Did this e.g. differ depending on the investigated species or duration of exposure?

3. Line 44-45: incomplete sentence

4. There is still one paragraph at the end of the introduction missing that clearly describes the

objectives and experimental setup of the current study (so far this is only described in two sentences in lines 75-76 and 89-90). Please clearly state the aim of this study at the end of the introduction.

Methods:

5. Line 365: Why did the water temperature in the two control treatments (CC and SC: 26 and 28.4 °C – which temperature belongs to which treatment?) differ so much? In this case, the larvae were not exposed to the same control conditions and it cannot be assumed that the temperature difference of more 2 °C did not affect the development of the larvae. Please give a reason for this large temperature difference, clearly state this information in the manuscript (not only in the methods section) and all figure captions and include this in the discussion of your results.
6. Line 369: You state a number of three replicates per treatment, however in Fig. 1 there are five replicates for each treatment. Please specify what is correct.
7. Line 384: You state in the rebuttal letter that you only analysed the lipid content of the larvae, however it is stated here that “We also measured the amount of lipids in the eggs according to a previously described method”. If this is the case, please give more information about the lipid analysis of the eggs, e.g. amount of eggs used? In addition, the cited reference (your own work) only measured lipid content of larvae, but not in eggs, right?
8. Line 392: What do you mean with “one hundred 5 days post-fertilization (dpf)”? Do you mean that you collected 100 5-day old larvae?
9. Line 400: If I understand it correctly, reference 44 did not measure the lipid content of larvae, but only of adult corals.
10. Line 421: How did you measure the length of the recruits if the pictures were taken from the top? What was used as a scale in the pictures?
11. Lines 437-440: You state that you used a GLM for the data that did not meet the assumptions of normality, but in this case it is not correct to use a Gaussian distribution for the GLM. Please re-analyse the data with the correct distribution of the data.
12. Line 138-139: Please specify which treatments you mean here.

Discussion:

13. Some parts of the discussion are still a repetition of the results, which can be deleted.
14. First paragraph (lines 163-171): You cite a lot of studies here that found different results, but what does this actually mean for the corals if the number and size of eggs changes under elevated temperatures? I think some more discussion on this aspect would be helpful for the readers (as you do in the third paragraph), maybe think about restructuring the first paragraphs.
15. You write that the lipid content was not measured in this study (line 198), but that the “lipid content of coral eggs was positively correlated with their survival rates” (line 205). Please make sure to omit all inconsistencies in the whole manuscript!

Figures:

1. Figure 2a: I would suggest to change the range of the Y-axis to 50-255 as this will make it clearer that none of the fragments bleached completely, but this is just a suggestion. Thank you for a better explanation of the colour scale in the text and for citation of the reference.
2. Figure 3c: As the two boxplots are overlapping, this does not look like a very significant difference between the larval size in the two treatments ($p < 0.001$ ***) and I would suggest to double-check the statistical analysis.
3. Supplementary Figure 1: You write in the rebuttal letter that no bleaching occurred but in Figure S1 you write “bleached fragment after stress”. Please clarify in the text and the figure if bleaching occurred or not.
4. Figure 2: You state that the survival rates are not significantly different, but $p = 0.03$ (line 679) and therefore $p < 0.05$.

Reviewers' comments:

Reviewer #1 (Remarks to the Author):

The authors have addressed most of the reviewer comments and suggestions by adding relevant context for their study in the introduction, and by discussing their results in comparison to findings described in previous studies. The methods section now has more detail, although some information is still lacking.

The paper is markedly improved content-wise and conclusions are appropriately stated. However, I recommend that the manuscript undergo extensive editing to improve clarity of the statements throughout and to reduce redundancy, especially in the introduction and discussion.

Specific comments:

Title

Line 1: "...life stages of the coral..."

Answer: we changed "stage" to "stages".

Intro

Line 41: "photosynthesis" instead of "photosynthetic activity"

Answer: Thank you for your suggestion. We changed "photosynthetic activity" to "photosynthesis" (Line 41)

Line 45: may bring what? sentence is incomplete

Answer: We apologize for the incomplete sentence. We changed here to " On the other hand, most *Acropora* spp. cannot survive¹³. The differences in the responses against the higher water temperature are due to species and durations of stresses." (Line 43-45)

Line 56-57: not clear what the relevance of life stage length is in this paragraph

Answer: Thank you for your advice. We add the duration of each stage. "For example,

larval stages are short (< less than two months), but the juvenile stages after settlement until reproduction are longer (> more than five years)." (Line 58-60)

Line 61-62: please rephrase for clarity

Answer: Thank you for your suggestion. We changed it to "In addition, high water temperature (29 to 32 °C) does not consistently cause negative impacts, such as mortality, on their larval stages in several species (mostly < 48 h)" (Line 63-65).

Line 63: "decreases" instead of "declines"

Answer: Thank you. We changed it to "decreases" (Line 66)

Line 76: is "consecutive" correct?

Answer: Yes, We changed this sentence as follows; "This study intended to reveal how consecutive stress on adult corals affects larval and juvenile performance under similar stress conditions" (Line 78-80)

Results

Line 94: First paragraph of results needs a short introduction to what was done to test physio responses of adult colony fragments

Answer: Thank you for your suggestion. We added the following sentence "First, we measured the brightness of the coral fragments, a color scale that corresponds to symbiont loss." (Line 105-106)

Line 108: "one fragment from the stressed treatment died 3 months after the thermal stress"

Answer: Thank you. We changed as you indicated (Line 120)

Discussion

Line 189-190: does this mean that bleaching was observed in the treatments for the adult colony fragments?

Answer: As indicated in the results, brightness increased after the thermal stress, but the apparent bleaching did not occur. We changed this sentence as follows; "In this study, we

applied a marginally shorter duration of heat stress, but apparent bleaching did not occur."
(Line 199-200)

Line 194-196: not sure what is meant by this statement

Answer: We would like to explain the colony changes its gametogenesis to increase its eggs in response to the thermal stress. Then we changed these sentences as follows; "Corals may abandon investing energy in gamete production when they are highly stressed, but they produce more eggs when they are slightly stressed. There may be a range of length and temperature of heatwaves, and we need to specify this range." (Line 205-207)

Line 220: statement on growth of marine invertebrates seems out of place in a paragraph discussing lipid utilization

Answer: Thank you for your suggestion. We erased this sentence.

Line 223: are there other studies where slower rates of lipid utilization were also observed at elevated temperature?

Answer: Thank you for your suggestions. We added sentences with reference "Lipid consumption rates in *P. damicornis* becomes slower in the higher water temperature⁵⁹" (Line 233-234).

Line 224: could the decrease in lipid utilization be because the larvae are not healthy at higher temperature?

Answer: We are also confused about why the lipid consumption rates decreased at the higher temperature. Physiological reactions become rapid according to the water temperature, but the lipid consumption rates become slower. We could not decide whether the larvae were healthy or not due to the water temperature here.

Line 225-226: what is meant by this statement?

Answer: We would like to describe as follows; "The lipid consumption rates change according to the water temperature, but further study is required to elucidate the relationship

between physiological function and the rate of lipid consumption rates." (Line 235-237)

Line 230-243: not sure what is meant by these statements; please clarify

Answer: We apologize for this statement. We changed these sentences as follows; "The size of the recruits to juveniles affects their survival rates. The larger size of the recruits shows higher survival rates in the brooding coral *Pocillopora damicornis* 40. The contradiction of the size and survivorship in this study, smaller recruits showing more heat tolerance, implies reactions to the thermal stress could be varied, and responses are under complex physiological reactions" (Line 241-245)

Line 235: "showed slighted higher heat tolerance"

Answer: Thank you for your suggestion. We changed "showed slight heat tolerance compared to those from non-stressed control" according to the other reviewer's comment (Line 246-247).

Line 236: rephrase for clarity

Answer: Thank you for your advice. We changed this sentence as follows; "their higher survival at larval to recruit stages in higher temperature are slightly beneficial for reef maintenance/resilience in the warming ocean" (Line 247-248).

Conclusion

Line 253: "...survival rates of larvae and recruits from control and stressed fragments..."

Answer: Thank you for your suggestion. We changed it as follows; "the survival rates of larvae and recruits from control and stressed fragments" (Line 265-266)

Line 255: what is meant by "parents may prepare eggs against heatwaves after the subsequent spawning"? please rephrase for clarity

Answer: Here, we mean that parents prepare eggs for larvae surviving in the heatwaves which could arise next year. Therefore, we changed this sentence as follows; if parents

experience the heatwaves, they may prepare eggs to survive their larvae in heat waves, which could arise next spawning (Line 270-271).

Methods

Line 402: were larvae at 21 days still swimming? Were these setups separate from the ones for settlement? was a settlement cue or substrate added into these setups?

Answer: We appreciate the reviewer's comments. They were swimming after 21 days, and we collected them from separated containers as larval stocks of each treatment without substrate. This sentence has been modified in the revised Ms." From each treatment (control and stress fragments), five replicates of 50 swimming larvae were randomly selected from larval stocks on the first and after 21 days of the experiment to obtain the total amount of lipid in larvae." (Line 416-417)

Line 410: how old were the larvae when they were placed in the settlement setups?

Answer: This was five days old larvae, and added "5 days post-fertilization (5dpf) "." (Line 425)

Line 417: were larvae maintained at high temperature for up to 2 months prior to measurement of size?

Answer: This point has been clarified in the revised Ms as "After 21 days of thermal stress experiment on recruits, recruits were maintained at ambient temperature in outdoor tanks with flow-through seawater under natural solar irradiation".(435-437)

Reviewer #2 (Remarks to the Author):

The authors have addressed my prior concerns through the addition of text within the introduction on the motivations behind the study and background literature in both the

introduction and discussion to provide context to their results. The text still requires polishing, especially the new text, before it can be published.

Below are my new comments on the revised manuscript.

Second Paragraph of Introduction: I found the two comparisons in this paragraph to be a bit confusing as currently written. What I think you are saying is, some corals like *Porites lutea* can withstand thermal stress unharmed but other species, like {fill in blank} cannot. They experience declines in physiological performance such as those referenced starting on Line 45.

Line 44-45: The sentence starting with "However, the higher..." is incomplete.

Answer: We apologize for the incomplete sentence. We changed this sentence into " On the other hand, most *Acropora* spp. cannot survive ¹³. The differences in the responses against the higher water temperature are due to species and durations of stresses." (Line 43-45)

Line 49: Do you mean adult reproductive output "as measured by" egg production varies?

Answer: Yes. Thank you for your suggestion. We added "as measured" (Line 51)

Line 49-50: I struggled through the first part of this sentence. Perhaps you might change it to, "The effect of heat stress on reproduction is often negative in many coral taxa" or a shorter version "For many coral taxa, the effect of heat stress reduces the number of eggs." When you refer to the eggs become smaller, are you talking about total number of eggs released or the size of eggs released decreases?

Answer: Thank you for your suggestion, and we apologize for its clarity. We mean here about a number, we changed "For many coral taxa, the effect of heat stress reduces the number of eggs" (Line 52)

Line 53: Change "between egg size and fecundity is" to "between egg size and fecundity has not been".

Answer: Thank you for your comment. Changed it to "has not been" (Line 55)

Line 53: I would consider changing "larval traits derived from stressed colonies" to "the inheritance of larval traits from stressed coral colonies".

Answer: Thank you for your advice. Changed it to "inheritance of larval traits from the stressed" (Line 55)

Line 62: Do you mean "larval survival" where you have written larval stages?

Answer: We changed from "In the larval stage" to "in terms of larval survival" (Line 62)

Line 68: I was a bit lost at the comparisons at the end of this sentence. Please revise.

Answer: We changed this sentence as follows; "Although the early life stages are crucial for the life cycles of the corals, the variations of the reactions against heat stress at early life stages are not consistently examined between larvae from stressed- and non-stressed adults." (Line 71-73)

Line 75: "This study intended.." (past tense).

Answer: Thank you for your comment. Changed it to past tense "intended" (Line 78)

Line 76: I would consider changing how this sentence is written. For example, "to reveal how consecutive heat stress on the adult corals effects the larval and juvenile performance under similar stress conditions."

Answer: Thank s again for your suggestion. We changed this sentence to "to reveal how consecutive stress on adult corals affects larval and juvenile performance under similar stress conditions."(Line 78-80)

Line 85: more heat "tolerant" larvae

Answer: Thank s again for your suggestion, Changed "tolerance" to "tolerant" (Line 89)

Line 94: Introduce brightness as a metric for bleaching. For example, "First we measured the brightness of the coral fragments, a color scale that corresponds to symbiont loss "

Answer: Thank you for your suggestion. We added the following sentence "First, we measured the brightness of the coral fragments, a color scale that corresponds to symbiont loss." (Line 105-106)

Line 114: The mean \pm standard error of the mean, SE can be removed here as it is also presented on Line 101.

Answer: Thank you for your suggestion. We removed "The mean \pm standard error of the mean, SE" (Line 126)

Line 140: I think you mean settlement size rather than larval size.

Answer: Thank you for your suggestion. We changed "larval size" to "settlement size" (Line 152)

Line 193: "Lepstastrea is not phylogenetically related to the genus Acropora,"

Answer: Thank again for the suggestion. We changed the position of not as you indicated. (Line 203)

Line 199: Is there a reference for this statement?

Answer: Not in references. We weakened this statement as follows; "the smaller lipid content of the larvae may represent the content of the lipid in the eggs." (Line 209-210)

Line 217: I am a bit confused here. You are unsure if any larvae in this experiment are symbiotic, so are you saying that this might be a strategy for survival in the field (outside the scope of this experiment). Please clarify.

Answer: Thank you for your suggestion, and we also apologize for confusing you. We mean here that the larvae from the stressed fragments had a smaller amount of the lipid, but they can build a symbiotic relationship in nature. The symbiotic relationship supports

energy supplement, and thus low lipid larvae can survive. "Although our study shows in cases of low lipids aposymbiotic larvae, the larvae could build symbiotic relationships in nature. Therefore, the survival rates of the low lipid larvae from SF parents could increase." (Line 226-228)

Line 235: Larvae and recruits from the stressed fragments "showed" slight heat tolerance "compared to" those from non-stressed control.

Answer: Thank you for your advice. We changed as you indicated. (Line 246)

Line 235: Only the SS recruits were sig. different from CS recruits, right? The larvae did not show differences in heat tolerance (no sig. differences).

Answer: Yes. In the control condition, recruits showed no significant difference in responses.

Line 246: juveniles

Answer: Thank you for your suggestion. We changed "juvenile" to "juveniles". (Line 258)

Reviewer #3 (Remarks to the Author):

Dear authors,

thank you for thoroughly revising your manuscript. The results are stated much clearer now, the introduction gives a better overview of the state-of-the-art, the methods are stated more clearly and it is much easier now to fully understand the experimental design and you discussed your findings in more detail. However, it is a pity in my opinion that no other environmental parameters than water temperature were measured during the experiment to ensure that only the temperature differed between the treatment conditions (both in experiment 1 and experiments 2-4). As all aquarium systems in experiment 1 were supplied with water flow-through, I assume that all other parameters were the same in both treatments in this experiment, but we cannot be sure for experiments 2-4 and I think it would be important to clearly state this limitation in the manuscript. I still noticed

some inconsistencies in the manuscript (see my detailed comments below) that should be remedied, in my opinion. I am particularly concerned that the temperature of the two control treatments (CC and SC) differs by more than 2 °C as such a large temperature difference will most likely affect the development of the larvae and it cannot be assumed that they were exposed to the same “control” conditions. Therefore, I would like the authors to give a reason why this temperature difference occurred, clearly state this information in the manuscript (not only in the methods section) and all figure captions and include this in the discussion of their results.

Answer: We set the ambient temperature in June for larval experiments. This month changes from the rainy season to high sunlight summer. Therefore, the water temperature changes drastically. We did not measure the other environmental parameters which are desirable, but we would appreciate it if you evaluated this study.

Abstract:

1. I do not agree with the strong statement that “heat-vulnerable corals have strategies to produce more heat-tolerant larvae after they experience stress” (lines 28-30) as your results show that the larvae of the control treatment were larger, had higher lipid content and had higher survival rates than the heat-stressed larvae (lines 121-139). Please clarify here.

Answer: Thank you for your suggestion. We changed this sentence to show that stressed corals produce slight heat-tolerant larvae in the "higher water temperature". "Corals produce more heat-resistant larvae in higher water temperatures after they experience stress." (Line 28-29).

Introduction:

2. Line 41-48: This paragraph is a bit confusing because you first state that corals might be able to acclimate to elevated temperature, but then you write that elevated temperatures will decline their physiological performance. Please state the state-of-the-art more clearly and why different studies found different responses. Did this e.g. differ depending on the investigated species or duration of exposure?

Answer: Thank you for your suggestion. We divided it into two paragraphs into this paragraph to show it more precisely. (Line 40-50)

3. Line 44-45: incomplete sentence

Answer: We apologize for the incomplete sentence. We changed here to " On the other hand, most *Acropora* spp. cannot survive ¹³. The differences in the responses against the higher water temperature are due to species and durations of stresses." (Line 43-45)

4. There is still one paragraph at the end of the introduction missing that clearly describes the objectives and experimental setup of the current study (so far this is only described in two sentences in lines 75-76 and 89-90). Please clearly state the aim of this study at the end of the introduction.

Answer: Thank you for your suggestion. We add a new paragraph to describe the objectives of this study (Line 93 - 101).

Methods:

5. Line 365: Why did the water temperature in the two control treatments (CC and SC: 26 and 28.4 °C – which temperature belongs to which treatment?) differ so much? In this case, the larvae were not exposed to the same control conditions and it cannot be assumed that the temperature difference of more 2 °C did not affect the development of the larvae. Please give a reason for this large temperature difference, clearly state this information in the manuscript (not only in the methods section) and all figure captions and include this in the discussion of your results.

Answer: This difference is due to changes in ambient temperature. The experiments were conducted in early summer, and thus water temperature changes rapidly. We explain this in methods as follows "The control conditions were ambient temperature in early summer. The water temperatures changed within a day and during experimental days." (Line 380-382). In the discussion, we also added sentences about this "We set ambient temperature as the control condition in the early summer; thus, the water temperature was not stable

and varied by about 2 °C (26 to 28 °C). However, the higher water temperature was stable within 1 °C difference (31 °C). There were at least 3-degree differences." (Line 267-270).

6. Line 369: You state a number of three replicates per treatment, however in Fig. 1 there are five replicates for each treatment. Please specify what is correct.

Answer: We apologize for the mistake. We did five replicates and edited the manuscript from three to five. (Line 384)

7. Line 384: You state in the rebuttal letter that you only analysed the lipid content of the larvae, however it is stated here that "We also measured the amount of lipids in the eggs according to a previously described method". If this is the case, please give more information about the lipid analysis of the eggs, e.g. amount of eggs used? In addition, the cited reference (your own work) only measured lipid content of larvae, but not in eggs, right?

Answer: We deeply apologize for making a mistake here. We did not measure the lipid amount of eggs and thus deleted this sentence.

8. Line 392: What do you mean with "one hundred 5 days post-fertilization (dpf)"? Do you mean that you collected 100 5-day old larvae?

Answer: Thank you for your comments, and we apologize to provoke your misunderstanding. Yes, here we mean 100 5 day old- 5 days post fertilization larvae. We changed "one hundred 5 days post fertilization (dpf) larvae were " (line 407)

9. Line 400: If I understand it correctly, reference 44 did not measure the lipid content of larvae, but only of adult corals.

Answer: Thank you for your comment. We made a mistake to cite the reference. We changed the reference to our previous paper, Hazrati et al., 2022 (55). (Line 414-415)

10. Line 421: How did you measure the length of the recruits if the pictures were taken from the top? What was used as a scale in the pictures?

Answer: Thanks for the reviewer's suggestions. We measured the longest diameter of the recruits and thus added "longest diameter (mm)" (Line 438)

11. Lines 437-440: You state that you used a GLM for the data that did not meet the assumptions of normality, but in this case it is not correct to use a Gaussian distribution for the GLM. Please re-analyse the data with the correct distribution of the data.

Answer: The statistics have been rechecked. For larval size, we used an unpaired two-sampled Wilcoxon test, and for the settlement rate between treatments, a pairwise Wilcoxon test was used instead of GLM. (Line 452-456)

12. Line 138-139: Please specify which treatments you mean here.

Answer: Here, we mean CC-CS and SC-SS (Line 151)

Discussion:

13. Some parts of the discussion are still a repetition of the results, which can be deleted.

Answer: Thank you for your advice. We erased several repeated statements of the results.

14. First paragraph (lines 163-171): You cite a lot of studies here that found different results, but what does this actually mean for the corals if the number and size of eggs changes under elevated temperatures? I think some more discussion on this aspect would be helpful for the readers (as you do in the third paragraph), maybe think about restructuring the first paragraphs.

Answer: Thank you for your advice. We erased and edited sentences to emphasize and summarize the results of this study.

15. You write that the lipid content was not measured in this study (line 198), but that the "lipid content of coral eggs was positively correlated with their survival rates" (line 205). Please make sure to omit all inconsistencies in the whole manuscript!

Answer: We apologize for the inconsistency. We did not measure the lipid contents and changed "coral eggs" to "lipid content of the larvae". (Line 210)

Figures:

1. Figure 2a: I would suggest to change the range of the Y-axis to 50-255 as this will make it clearer that none of the fragments bleached completely, but this is just a suggestion. Thank you for a better explanation of the colour scale in the text and for citation of the reference.

Answer: We appreciate the reviewer's suggestions. The figure has been modified following the suggestions of the reviewer.

2. Figure 3c: As the two boxplots are overlapping, this does not look like a very significant difference between the larval size in the two treatments ($p < 0.001$ ***) and I would suggest to double-check the statistical analysis.

Answer: This point has been double-checked and clarified in the revised Ms. The result has significantly differed between treatments.

3. Supplementary Figure 1: You write in the rebuttal letter that no bleaching occurred but in Figure S1 you write "bleached fragment after stress". Please clarify in the text and the figure if bleaching occurred or not.

Answer: Thanks for the reviewer's comment. Bleaching did not occur. This point has been modified in Figure S1.

4. Figure 2: You state that the survival rates are not significantly different, but $p = 0.03$ (line 679) and therefore $p < 0.05$.

Answer: We appreciate the reviewer's comments. This mistake has been corrected in the revised Ms. "Coral fragments in the two temperature treatment groups were not significantly different survival rates ($P=0.03$)" (Line 689-690)